# Prion Mutations in Republic of Republic of Korea, China, and Japan

**DOI:** 10.3390/ijms24010625

**Published:** 2022-12-30

**Authors:** Dan Yeong Kim, Kyu Hwan Shim, Eva Bagyinszky, Seong Soo A. An

**Affiliations:** 1Department of Bionano Technology, Gachon University, Seongnam 13120, Republic of Korea; 2Department of Industrial and Environmental Engineering, Graduate School of Environment, Gachon University, Seongnam 13120, Republic of Korea

**Keywords:** prion, Creutzfeldt–Jakob disease (CJD), fatal familial insomnia (FFI), Gerstmann–Sträussler–Scheinker disease (GSS), mutation, risk modifiers

## Abstract

Prion gene (PRNP) mutations are associated with diverse disease phenotypes, including familiar Creutzfeldt–Jakob Disease (CJD), Gerstmann–Sträussler–Scheinker disease (GSS), and fatal familial insomnia (FFI). Interestingly, PRNP mutations have been reported in patients diagnosed with Alzheimer’s disease, dementia with Lewy bodies, Parkinson’s disease, and frontotemporal dementia. In this review, we describe prion mutations in Asian countries, including Republic of Republic of Korea, China, and Japan. Clinical phenotypes and imaging data related to these mutations have also been introduced in detail. Several prion mutations are specific to Asians and have rarely been reported in countries outside Asia. For example, PRNP V180I and M232R, which are rare in other countries, are frequently detected in Republic of Korea and Japan. PRNP T188K is common in China, and E200K is significantly more common among Libyan Jews in Israel. The A117V mutation has not been detected in any Asian population, although it is commonly reported among European GSS patients. In addition, V210I or octapeptide insertion is common among European CJD patients, but relatively rare among Asian patients. The reason for these differences may be geographical or ethical isolation. In terms of clinical phenotypes, V180I, P102L, and E200K present diverse clinical symptoms with disease duration, which could be due to other genetic and environmental influences. For example, rs189305274 in the ACO1 gene may be associated with neuroprotective effects in cases of V180I mutation, leading to longer disease survival. Additional neuroprotective variants may be possible in cases featuring the E200K mutation, such as KLKB1, KARS, NRXN2, LAMA3, or CYP4X1. E219K has been suggested to modify the disease course in cases featuring the P102L mutation, as it may result in the absence of prion protein-positive plaques in tissue stained with Congo red. However, these studies analyzed only a few patients and may be too preliminary. The findings need to be verified in studies with larger sample sizes or in other populations. It would be interesting to probe additional genetic factors that cause disease progression or act as neuroprotective factors. Further studies are needed on genetic modifiers working with prions and alterations from mutations.

## 1. Introduction

Prion diseases are neurodegenerative diseases caused by neurotoxicity due to the accumulation of abnormally folded prion proteins. Prion diseases have been reported in humans, as well as in different animals, such as hamsters, voles, mice, minks, felines, bovines, sheep, deer, elks, and goats. [1,2,3]. Normally, cell-surface prion protein (PrP^c^) has important diverse roles, including neural protection, cell adhesion, synaptic connections, neurotransmission, and cell signaling. The PrP^C^ contain approximately 66% a helices with a low proportion of b sheets [1,4]. The PrP^C^ may convert to a misfolded isoform (PrP^Sc^), which is rich in b sheets and resistant to proteases. PrP^Sc^ mediates conversion of PrP^c^ to PrP^Sc^. PrP^Sc^ can propagate and accumulate in the central nervous system, leading to neurodegeneration [1].

A small percentage (10–15%) of prion diseases may be related to genetic mutations in the prion gene (*PRNP*). *PRNP* is located on chromosome 20. The gene contains two exons. The first is a non-translating exon that contains the 5′-untranslated region (UTR) leader. Exon 2 includes the protein coding sequence and the 3′ UTR region. PrP^c^ is 253 amino acids in length. Two post-translational cleavages of PrP^C^ have been identified in the prion protein; they feature the removal of the first 22 amino acids in the N-terminal region and the last 23 amino acids in the C-terminal region. The C-terminal region, the glycophosphatidyl inositol (GPI) sequence, anchors prion proteins to cell membranes [5]. The PrP^C^ protein contains three a helices, two small b sheet regions, a long N-terminal loop, and a short C-terminal loop [1]. The N-terminal area of prion protein contains five octapeptide repeats of the P(H/Q) GGG(G)WGQ sequence. The removal of one repeat may not result in any disease phenotype. However, insertions in several octapeptide sequences (OPRI) have been associated with prion diseases. [5]. A disulfide bond is formed between C179 and C214, located in helix 2 and 3, respectively. These cysteines are highly conserved and the disulfide bond plays a critical role in prion folding and stability [6]. Conserved glycine- and alanine-rich regions are located between A113 and Y128. This sequence may play a significant role in prion function and stability [5]. Two glycosylation sites were identified in PrPC (N181 and N197), which are occasionally occupied by complex N-glycans. These residues were suggested to play a crucial role in neuroprotection and in prevention of protein assembly and toxicity [7]. Figure 1 shows a schematic structure of prion protein with important positions and residues. The polymorph methionine 129 and glutamic acid 219 residues are also included.

Genetic prion diseases can have diverse phenotypes, including Creutzfeldt–Jakob disease (CJD), fatal familial insomnia (FFI), and Gerstmann–Sträussler–Scheinker disease (GSS) [8]. Typical familial CJD is associated with rapid disease progression, short survival time (less than a year), and progressive dementia with motor dysfunctions (myoclonus, tremor). In the brain, PrP^Sc^ plaques may be associated with gliosis and neuronal loss. The initial symptom of GSS may be ataxia, or Parkinsonism (such as tremor, bradykinesia, rigidity, and postural instability) and dementia may appear later in life. The disease duration may be variable; several GSS patients may die within a year, while the majority of patients may survive for several (even more than 10) years. Amyloid plaques, which contain amyloid beta (Ab) peptide aggregates, may appear in the brain, particularly in the cerebellum. The initial symptoms of FFI usually include insomnia and dysautonomia, followed by motor and cognitive impairments in later disease stages. Disease duration may be relatively short; patients can die less than 2 years after disease onset. FFI neuropathy may be diverse and includes loss of thalamic nerves, thalamic atrophy, inferior olivary nucleus atrophy, or PrP^Sc^ deposition in the midbrain or hypothalamus [9]. All three genetic prion diseases may represent atypical forms of disease with different symptoms and longer or shorter disease durations. Furthermore, atypical disease phenotypes may also be related to genetic prion mutations. Several patients with prion mutations were diagnosed with Alzheimer’s disease (AD), frontotemporal dementia (FTD), and dementia with Lewy bodies (DLB). Similarities have been observed between prion diseases and other neurodegenerative diseases (AD and FTD), as all of these diseases are associated with misfolded protein aggregation (such as Ab, microtubule associated Tau protein). Also, PrP^Sc^ may accumulate together with Ab peptides or abnormally folded tau protein [1]. Besides the common prion disease-related variants in Asian patients (such as P102L, V180I, E200K, M232R), several unique rare variants have appeared in Korean, Chinese, and Japanese patients, which were reported only in a single patient (will be discussed later). The majority of rare mutations may not have strong evidence of pathogenicity, especially if they do not have any family history of disease or segregation cannot be proven [10].

In this review, *PRNP* mutations discovered in Asian patients (especially in Republic of Korea, China, and Japan), are introduced with descriptions of related clinical phenotypes and disease courses. Currently, the impact of the disease from prion mutations in Asia was extensively studied only in Republic of Korea, China, and Japan. Neighboring countries also revealed diverse prion mutation patterns, which would be discussed for their possible disease ramifications by understanding the differences, as well as in comparison to other non-Asian populations. Furthermore, a few putative genetic disease modifier factors were also observed in these patients with prion mutations. We also will discuss the possible genetic modifier factors described in Asian patients.

## 2. Prion Mutations in Republic of Korea

In Republic of Korea, the different disease-related variants (such as V180I, D178N, E200K, or M232R) were reported in patients diagnosed with CJD, FFI, or GSS. The possible risk modifiers (M129V and E219K) in the prion gene were also analyzed [10,11,12,13,14,15,16,17,18,19,20,21,22,23,24,25,26,27,28,29,30,31,32]. M129V and E219K were reported as variants with uncertain significance. Conflicting reports are available on them; they were suggested either as risk modifiers or risk factors for prion diseases, but other studies refuted their association with neurodegenerative diseases. In East Asia, E219K may protect against sporadic CJD (sCJD), as heterozygous E219K was relatively common among healthy Japanese individuals (7%). However, it may be rare among Caucasians. Both heterozygous E219K and M129V could possibly impact variant CJD (vCJD), especially in Caucasians [1,11]. Jeong et al. published the first Korean study on prion mutations in 2004 [11]. Frequencies of M129V and E219K were analyzed in healthy Koreans. The M129 analysis revealed 94,33%, 5.48%, and 0.19% for the M/M, M/V, and V/V genotypes, respectively. The homozygous methionine allele was similar, compared to the studies in Japan, but it was higher, compared to the normal UK population. The E219 genotyping revealed 92.06% and 7.94% for E/E and E/K alleles, respectively (K/K allele was not observed). The frequency was higher than in the Japanese population. This study suggested that the MM genotype of M129V and the EK genotype of E219K may be specific to East Asian populations. A single octapeptide deletion was detected in two unaffected Korean individuals; however, it may not be associated with any form of disease. The authors also compared the frequencies of normal variants between Korean and European individuals. The R2 deletion was detected only among Koreans, whereas normal Europeans could harbor R2-R3 or R3-R4 deletions. Furthermore, several possible benign missense or silent variants, which were described in Europe, such as D171S, A117A, G124G, and V161V, were missing in the Korean general population. These different mutational patterns between Koreans and Europeans suggest that there may be ethnicity-specific variants that may be prominent in Europe, but missing (or rare) in Asia [11]. Another study by Jeong et al. (2005) examined the allelic distribution of M129V and E219K in 150 Korean sCJD patients. None of the patients carried any of these variants. This study may support the suggestions that these variants could protect against sCJD in East Asia [12].

A 2007 paper by Jeong et al. [13] analyzed M129V and E219K in AD patients. The authors compared 276 patients with sporadic AD to 236 unaffected Korean individuals. No significant differences in the genotypes of codons 129/219 or their haplotypes were evident. These data suggest that codon 129/219 variants may not be directly associated with sporadic AD in Republic of Korea [13]. Similar data were detected among Japanese patients before [14]. 

The N97S mutation was reported in 2012 while analyzing the suspected CJD group and the Korea Association Resource group (KARE), which suggested it as a possible benign variant. A KARE cohort study was established by Korea National Institutes of Health (KNIH), and they performed genome-wide association studies (GWAS) on a large community cohort. The goal of this study was to find genetic risk factors for diseases. The study by Lee et al. (2012) analyzed 22 patients with definite prion disease, 163 patients with suspected prion disease, and 296 individuals from the KARE group (they were randomly selected) [15]. No detailed reports are available on the carrier’s disease phenotype, symptoms, or age of onset. The KARE study also revealed two silent variants (P68P and N197N) and 11 different variants in the promoter region, including rs77420351, rs2756271, or rs57633656. This study failed to find any association between non-coding variants and neurodegenerative diseases. Limitations of this study were the insufficient epidemiological results and lack of detailed clinical data [12]. The KARE study analyzed the frequency of the M129V and E219K variants in the three different groups. In the definite prion disease group, the M/M and M/V allele ratio was 91% and 9%, respectively. The E/E and E/K allele ratios (for codon 219) were 94.5 and 5.5, respectively. Among the suspected CJD patients, the frequency of the M/V and E/K alleles were 4.91% and 6.79, respectively. In the KARE group, these respective frequencies of the M/V and E/K alleles were 5.8% and 7.82%, respectively. Furthermore, in the KARE group, one homozygous E219K mutation in the K/K allele was observed; this mutation may be very rare. This study found lower 129M/M and 219E/E allele frequencies in patients, compared to the studies by Jeong et al. (2005) [13,15]. However, the sample size of these studies was too small for precise analysis of these variants [15]. A recent study by Kim and Jeong (2021) performed meta-analysis on M129V variation and its association with sCJD. They found that the homozygous MM genotype may be a possible risk factor for sCJD, compared to the heterozygous MV genotype. They also suggest that similar analysis may be needed in case of the association between E219K and sCJD [16]. 

Besides probable nonpathogenic variants and risk modifiers, pathogenic or probable disease-associated mutations have also been reported in Republic of Korea. These include P102L, D178N, V180I, E200K, and M232R. The first Korean patient with the P102L mutation was described in 2010 by Park et al., which was also the first Korean case of GSS [17]. The 46-year-old female patient presented with slow progressive ataxic gait, language impairment, and cognitive dysfunction. Magnetic resonance imaging (MRI) revealed a strong intensity in the cerebral cortices. No mutations were detected in spinocerebellar ataxia-related (SCA) genes. The patient was positive for the 14-3-3 protein (14-3-3 is a signaling protein, CSF marker of prion diseases, including CJD), and the electroencephalography (EEG) signal revealed atypical non-specific slow waves. [17]. P102L was observed in one patient from the 2012 KARE study, who was diagnosed with definite GSS. However, no details regarding the clinical symptoms have been described [15]. In 2019, P102L al. was described in a patient with Kang et al. [18]. The patient presented with clinical phenotypes at 49 years of age, without any family history. Symptoms included progressive gait disturbance, slurred speech, and hand clumsiness. Memory dysfunction was also observed in the patient. MRI revealed high signal intensities in the bilateral cortices and mild cerebellar atrophy. EEG was normal, but the cerebrospinal fluid (CSF) was positive for 14-3-3 protein [18]. A recent (2022) report by Ahn et al. [19] described the first Korean familial case of GSS with P102L mutation. Phenotypic heterogeneity was observed in this family, as several family members were misdiagnosed with hereditary cerebellar ataxia. The proband was in her 40s and developed progressive gait disturbances. Later, she developed personality changes and rapidly progressive memory dysfunction. No mutations were observed in ataxia-related genes or huntingtin (HTT) gene. Diffusion-weighted imaging (DWI) showed high signal intensity in the hemispheric and caudate nuclei. EEG revealed mild diffuse slowing, and CSF 14-3-3 was positive. The brother of the proband carried the same mutations and developed ataxia, but no cognitive dysfunction. MRI did not reveal any abnormalities in him [19]. 

The D178N mutation (with homozygous M/M at codon 129) was first discovered in 2009 by Choi et al. [20] in a 67-year-old male patient with atypical CJD, without any family history. Symptoms included progressive gait disturbance and dysarthria with extrapyramidal signs, without insomnia. The patient also showed rigidity and bradykinesia, but no myoclonus, visual dysfunction, cognitive decline, or pyramidal symptoms. The patient’s condition was worsening rapidly into akinetic mutism in a few months, but disease duration was relatively long (more than 2 years). EEG results were normal, and the 14-3-3 CSF was positive. MRI revealed high signal intensities in both the parietal and occipital gyri. [20]. Another patient with D178N (M/M at residue 129) was described by Lee et al. in 2014 [21]. The 34-year-old male patient was diagnosed with FFI. Nine months prior to the admission to medical facility, the patient had sleep disturbances and abnormal breathing during sleep. He also experienced excessive sweating, tremors, and became restless. His executive functions and memory started to deteriorate 2 months before admission. A few days before admission, additional mobile impairment appeared in him, such as gait dysfunctions or instability in posture. Fluorodeoxyglucose-positron emission tomography (FDG-PET) revealed mild hypometabolism in the bilateral frontal cortices and bilateral thalamus. This patient was the first one, who was diagnosed with FFI in Republic of Korea. [21]. The third case of D178N was described in a 57-year-old male patient, diagnosed with FFI [22]. The patient experienced memory dysfunctions and sleep issues 5 months before being admitted to the medical facility. Symptoms included an irregular pattern of the sleep-wake cycle, visual hallucinations, myoclonus, ataxic gait, and weight loss. During the hospital visit, he was disoriented. Sleep disturbances became worse quickly. Abnormal eye, chin, and leg movements were observed during sleep. The sleep disturbance was also associated with motor hyperactivation (or agrypnia excitata). CSF 14-3-3 was negative, and EEG showed diffuse slowing without periodic discharge. MRI and PET revealed white matter lesions and lower uptake in the bilateral thalamus, respectively. [22]. A recent publication on PRNP D178N (M/M at codon 129) [23] reported a case of Parkinsonism with dementia (PDD). The patient was a 68-years-of-age female and presented with gait disturbances. Disease started with gait dysfunctions and regular falls. Neurological data revealed moderate bradykinesia, mild rigidity, mild dysphagia, and moderate postural instability. Although the patient had sleep disturbances (such as apnea), insomnia was not present. Myoclonus and ataxia were absent. The patient became bedridden in 7 months after first symptoms, and died after 15 months after hospital admission. MRI revealed mild leukoaraiosis. F-N-(3-fluoropropyl)-2beta-carbomethoxy-3beta-(4-iodophenyl)nortropane. (CIT)-PET revealed hypometabolism in the midbrain. The family history was positive; the patient’s mother and sister also presented with similar symptoms in their 60s. Additional atypical symptoms appeared in the sister, such as urinary dysfunction, breathing disturbances (inspiratory stridor, snoring), and abnormal sleep behavior. Both the proband and her sister presented rapidly progressive disease with several autonomic dysfunctions. The final diagnosis in this family may be considered hereditary prion disease [23].

The first case of PRNP V180I in Republic of Korea was observed in 2010 in an fCJD patient, who developed the disease at 75 years of age [24]. The disease started with depression, paranoia, hallucinations, and suicidal thoughts. Antidepressants were not effective. Four months after hospital admission, the patient developed progressive dementia and behavioral symptoms (such as stereotypic behavior), which became more prominent in 2 months. Bradykinesia, mild progressive parkinsonism, cerebellar ataxia, and upper-limb clumsiness subsequently appeared. EEG revealed slow waves in the right hemisphere, and 14-3-3 in the CSF was positive. MRI revealed high signal intensities in the bilateral frontal, parietal, temporal, and occipital cortices [24]. V180I was also described in a CJD study by the KARE group in three patients with suspected CJD [15]. As symptoms, dementia and ataxia were described, but no further details are included in the manuscript. Two additional individuals from the KARE group carried V180I, but they did not present any neurodegenerative phenotypes [15]. In 2013, Yeo et al. [29] reported an atypical CJD case of V180I. The patient developed disease symptoms at 75 years of age, without any family history. No detailed description was written on her clinical symptoms, but the patient was in semi-comatose stage at the time of hospital admission. MRI revealed high-intensity lesions in the thalamus, right frontal cortex, and right temporal cortex. EEG showed slowness in the background rhythm and periodic sharp wave complexes in the cerebral hemisphere. The CSF was also positive. Brain tissue analysis revealed spongiform changes, vacuolation, gliosis, and neuronal loss in most cerebral cortices, with the exception of the brainstem and thalamus. The atypical form of protein K-resistant PrP was detected by western blotting using 3F4 monoclonal antibody (mAb) [29]. A 2016 study by the National Institute of Health, Korea Centers for Disease Control and Prevention analyzed risk modifier variants in five patients harboring V180I and diagnosed with familial CJD (fCJD) [26]. Patients developed the disease between 57 and 73 years of age. Disease progression was slow in some patients and fast in others. The symptoms included cognitive dysfunction, ataxia, depression, or encephalopathy. This study revealed that additional mutations in other disease-related genes may act as disease-modifying factors in CJD patients with V180I mutations. Genes involved in this study included AD risk genes (for example Aconitase 1 or ACO1; Lipoprotein A or LPA; Periostin or POSTN; Structural maintenance of chromosomes protein 5 or SMC5) or PD risk genes (Fibroblast Growth Factor 20 or FGF20; leucine rich repeat kinase 2 or LRRK2; 2-Hydroxyacyl-CoA Lyase 1 or HACL1). Several variants were suspected to affect the prion disease course, including variants in the LPA, ACO1, FGF20, POSTN, or LRRK2 genes [26]. In 2019, an atypical case of CJD was reported in a 78-year-old male patient. His symptoms included visual hallucinations and anxiety, which started 5 months before hospital visit, followed by cognitive dysfunctions. Mobility issues, such as ataxia or pyramidal/extrapyramidal signs, were not observed in him. EEG was normal and CSF 14-3-3 signal was positive, but RT-QUIC for PrPSc was negative. MRI showed bilateral high intensities in several brain areas, including the frontal, temporal, and occipital cortex [27]. In 2019, V180I was reported in a 58-year-old female diagnosed with early onset AD (EOAD) [28], based on the National Institute on Aging-Alzheimer’s Association (NIA–AA) criteria. The patient showed cognitive decline and visuospatial dysfunction and had a positive 14-3-3 signal. The patient was also positive for AD markers, including elevated tau and Ab levels in CSF. It is possible that prions interact with other AD risk genes, resulting in AD progression [28]. The most recent publication on PRNP V180I [29] discussed CJD patients who developed the disease at 57 years of age. This patient is the longest survivor of CJD, with the disease duration of 16.5 years. The first symptoms were headache and anxiety in her late 50s. Six months later, the patient experienced rapid cognitive decline, depression, loss of appetite, and left-hand tremors. In 14 months after the first symptoms, her condition became worse and she developed akinetic mutism and myoclonic movement. She survived on life support (tube feeding and respiratory assistance) for an atypically long time. MRI revealed hyperintensities in the right basal ganglia and bilateral frontotemporal cortices. No atrophy was observed. The patient had a positive 14-3-3 signal in the CSF and periodic sharp wave complexes. Spongiform changes with neuronal loss were observed in the frontal cortex [29].

The first case of E200K in Republic of Korea was discovered in 2009 in a 58-year-old male patient. The patient also harbored a heterozygous M129V mutation. The patient showed rapid disease progression, gait dysfunction, confusion, and myoclonus. Shortly, dysarthria, reduced gait, reduced attention, and agitation also appeared. During the neuropsychological test, he presented confusion and disorientation. The patient died 3 months after the first symptoms. MRI revealed high signal intensity in different brain areas, including the bilateral frontal temporoparietal area and caudate nucleus. EEG revealed sharp spikes and slow waves, and CSF was positive for the 14-3-3 signal [20]. This mutation was described in a 2012 study by KARE. Two probable CJD patients were reported, but no details were provided regarding the clinical symptoms or age of onset [15]. A family with fCJD was reported in 2014. The proband patient developed progressive dysarthria and visual hallucination at the age of 63. In a short time, she also experienced gait disturbances, myoclonus, and behavioral issues. Her DWI showed high signal intensities in basal ganglia, occipitoparietal cortex. EEG revealed PSWs, and CSF 14-3-3 was positive. An 84-year-old non-CJD family member also carried the E200K mutation. Additional unrelated patients with E200K and controls were also included. This study identified possible protective factors against CJD [30,31].

The first CJD patient with PRNP V203I was reported in 2010 [32]. The patient also harbored a heterozygous M129V mutation. The patient was a 66-year-old female patient. She developed gait disturbances, rapid progressive decline, tremor, mild bradykinesia, and incontinence. Later, she presented myoclonic jerks, and mental and neurological symptoms worsened. MRI was initially normal, but later showed gyriform hyperintensity in the cerebral cortex [28]. This mutation was also mentioned in a study by KARE in a probable CJD patient, but no clinical details were reported [15].

The novel PRNP Y225C was reported in an atypical CJD case in 2019 [33] in a Korean patient. The patient had slow progressive memory decline and speech disturbances in his 50s. Rigidity and myoclonic jerk were also observed. In the late disease stage, patient became bedridden, had myoclonus spontoons eye opening, and made unintelligible screams. EEG revealed diffuse slow continuous delta activity in the bilateral cerebral hemispheres. CSF displayed a weak positive signal for 14-3-3. Tau was elevated in CSF, while amyloid peptides were reduced. MRI revealed high signal intensity in the cerebral cortex of the bilateral basal ganglia, frontal lobe, parietal lobe, and parietal lobe. FDG-PET showed decreased metabolism in the cerebral cortices, but not in the primary sensory motor cortex and occipital lobe. The family history remained unclear. Mutations may be associated with incomplete penetrance, but their pathogenic nature should not be ruled out [33]. 

The first case of M232R was reported in Republic of Korea in 2009, along with E200K and V203I mutations [20]. The patient was a 65-year-old male. There was no family history of dementia. He developed rapid progressive memory dysfunction and gait disturbances. The duration of the disease was 16 months. MRI showed high signal intensities in the parieto-occipital cortex and temporal lobes. EEG revealed diffuse theta to delta range slow waves. CSF was positive for 14-3-3 [20]. This mutation also was mentioned in a 2012 KCDC-KARE study [20]. One probable CJD patient with M232R developed CJD with ataxia, myoclonus, and mutism, but no further details were mentioned regarding clinical symptoms. Additionally, three individuals with M232R in the KARE did not develop neurodegenerative phenotypes. Follow-up studies may be needed that compare CJD patients and asymptomatic individuals with V180I or M232R mutations to verify the disease-modifying or neuroprotective factors [15]. PRNP M232R was also reported in a 73-year-old patient suspected to have corticobasal syndrome [34]. The diagnosis was later revised to fCJD. MRI showed gyriform high signal intensities in the frontal cortex, insula, bilateral parietal lobes, perirolandic gyrus, and occipitotemporal areas. Symptoms included gait disturbance, parkinsonism with bradykinesia, tremor, rigidity, stuttering, and swallowing difficulty. Motor impairments, such as parkinsonism, worsened rapidly. EEG revealed slow waves, and the 14-3-3 signal was weakly positive. Additionally, tau was elevated and amyloid peptides were reduced in CSF [30]. In 2017, M232R was described in a 62-year-old patient who developed CJD with an AD-like, slow progressive disease course [35]. Spasticity and cognitive dysfunction are the main hallmarks of this disease, followed by intermittent myoclonus. Disease duration was relatively long (at least 4 years). No family members with similar disease phenotypes were found, but the patient’s children carried the same mutation. EEG revealed a reduced signal, but no sharp waves were observed. MRI showed atrophy in the cortex of the frontal and cortical brain regions, and FDG-PET showed hypometabolism on both sides of the frontal, parietal, and temporal lobes. CSF 14-3-3 expression was weakly positive [35]. The most recent case of M232R was described in 2019 [36]. The patient was diagnosed fCJD at 57 years of age. Initial symptom was rapid progressive dementia. Motor functions remained normal, but 17 months after onset, myoclonus appeared. FDG-PET revealed severe glucose hypometabolism in the bilateral temporoparietal lobes and thalamus, but DWI did not reveal any abnormalities. EEG showed intermittent slow waves in the bilateral hemispheres. CSF was positive for 14-3-3, and RT-Quic was positive for PrP^Sc^ [36]. The findings are summarized in Table 1 and Figure 2a.

**Table 1 ijms-24-00625-t001:** Prion mutations found in Korean patients (AOO means age at onset, EEG means electroencephalography).

Mutation	Disease	AOO (Years)	Family History	Phenotype	14-3-3	EEG	Imaging	Remarks	Reference
N97S	NA	NA	NA	NA	NA	NA	NA	Probable benign	[15]
octapeptide deletions	NA	NA	NA	NA	NA	NA	NA	Probable benign	[10]
P102L	GSS	NA	NA	GSS	NA	NA	NA	NA	[15]
	GSS	49	−	gait disturbance, slurred speech	+	Normal	MRI: high signal at several brain areas	No SCA mutations	[17]
	GSS	40s-early 50s	+	Slowly progressive cerebellar ataxia	+	Mild diffuse slowing	MRI: high signal in hemispheric cortex and caudate nuclei	CSF-Tau +	[18]
	GSS	46	+	Severe dementia and dysarthria	+	slow waves	MRI: high signal in hemispheric cortex	No SCA mutations	[19]
M129V	NA	NA	NA	NA	NA	NA	NA	Risk modifier	[11,12,13,14,15,16]
D178N	CJD-129MM	67	−	Gait disturbance, dysarthria	+	Normal	MRI: High signal in parietal and occipital gyri	NA	[20]
	FFI-129MM	34	+	insomnia, dementia autonomic disturbances	−	NA	FDG-PET: hypometabolism in the midbrain hypothalamus	NA	[21]
	FFI-129MM	57	NA	Memory dysfunctions, sleep disturbance	−	NA	FDG-PET: lower uptake in bilateral thalamus	Agrypnia excitata	[22]
	PDD -129MM	68	+	Parkinsonism, dementia, no insomnia	−	NA	PET: hypometabolism of midbrain	No amyloid deposition	[23]
V180I	CJD	75	+	Neuropsychiatric symptoms, dementia	+	Slow waves	MRI: high signal intensities in different brain areas	NA	[24]
	CJD	NA	NA	Dementia, ataxia	NA	NA	NA	NA	[15]
	CJD	75	−	Atypical form, slower disease progression	−	PSWC	DWI: high signal in cerebral cortex, thalamus	spongiform encephalopathy	[25]
	CJD	57–77	NA	5 cases of CJD, rapid or slow progressive	+	Slow waves	DWI or MRI positive	3/5 were tau positive	[26]
	CJD	78	+	Dementia, visual symptoms, ataxia	+	Normal	MRI: high signal in frontal, parietal, temporal, occipital cortex	High Tau, lower amyloid in CSF	[27]
	EOAD	58	−	Memory and visuospatial dysfunctions	+	Normal	PET: mild amyloid positivity, MRI: hippocampal atrophy	High Tau, lower amyloid in CSF	[28]
	CJD	58	−	Cognitive dysfunctions, depression, tremor	+	Normal	MRI: hyperintensities in right different brain areas, no atrophy	Longest CJD survivor	[29]
E200K	CJD-129MV	58	−	Gait, confusion, agitation, myoclonus	+	Sharp or slow waves	MRI: High signal in bilateral front temporoparietal area and caudate nucleus	NA	[20]
	CJD	NA	NA	NA	NA	NA	NA	NA	[15]
	CJD	60–70s	+	Myoclonus, behavioral issues	+	PSWC	MRI: high signal in basal ganglia and occipitoparietal cortex	Possible neuroprotective genetic factors	[30,31]
	unaffected carrier	85		NA	NA	NA	NA		
V203I	CJD-129MV	66	−	Gait disturbance, cognitive dysfunction, myoclonus	+	NA	MRI: gyriform hyperintensity in cerebral cortex	NA	[32]
	CJD	NA	NA	NA	NA	NA	NA	NA	[15]
E219K	NA	NA	NA	NA	NA	NA	NA	Risk modifier	[11,12,13,14,15,16]
Y225C	CJD	54	NA	Slow disease progression, behavioral dysfunctions	+	Diffuse slow d activity	MRI: high signal in bilateral basal ganglia, frontal lobe, parietal lobe, and parietal lobe	NA	[33]
M232R	CJD	65	−	Progressive memory dysfunctions, gait disturbance	+	Diffuse t to d range slow waves	High signal in cortex of parieto-occipital and temporal lobes	NA	[20]
	CJD	NA	NA	Ataxia, myoclonus and mutism	NA	NA	NA	NA	[15]
	CJD	73	+	CJD, presented as corticobasal syndrome (CBS)	+	Slow waves	MRI: high signal in different brain areas	High Tau, lower amyloid in CSF	[34]
	CJD	60	−	AD-like symptoms, gait disturbance	+	Background activity reduced	FDG-PET: hypometabolism in frontal, parietal and temporal lobes	NA	[35]
	CJD	57	NA	Rapidly progressive dementia	+	Slow waves	MRI: diffuse cortical atrophy	CSF positive for PrPsc	[36]

## 3. Prion Mutations in Japan

In Japan, prion mutations were thoroughly studied. Since Republic of Korea and Japan were relatively close neighboring countries, similar mutation patterns were discovered, for example, P102L, V180I or M232R. Interestingly, differences were also reported, especially with P105L mutation, which was observed relatively commonly in Japanese patients, but not in Republic of Korea or China [37,38,39,40,41,42,43,44,45,46,47,48,49,50,51,52,53,54,55,56,57,58,59,60,61,62,63,64,65,66,67,68,69,70,71,72,73,74,75,76,77,78,79,80,81,82,83,84,85,86,87,88]. Several prion gene pathogenic mutations have been discovered and associated with CJD, GSS, FFI, or other disease phenotypes (Table 2, Figure 2b). Insertions near the N-terminal region have been reported in patients with CJD and familial prion disease. A 144-base insertion was found in 1995 in a case of early onset familial prion disease, in which the affected individuals presented with slow progressive cortical dementia and ataxia [37]. Schizophrenia-like symptoms or psychiatric dysfunction are also common. One family member was initially diagnosed with Pick’s disease. Movement impairment may be prominent (tremor, gait instability, extrapyramidal signs, and cerebellar ataxia). All patients developed disease at a relatively young age (late 20s or 30s). PrP-reactive plaques appeared in the brain, and EEG showed either slow waves or diffuse alpha pattern [37]. Another paper [38] described a four octapeptide-repeat insertion (OPRI) mutation associated with sporadic CJD, in which the patient had rapidly progressive dementia with cerebellar ataxia at the age of 56. Symptoms started with backpain and fatigue, which he developed 3 months before hospital admission. It was followed by visual impairment and gait disturbance shortly. He was diagnosed with dysarthria and lost the ability to walk. Later, he developed a speaking impairment, myoclonus dysphagia. The patient died in 5 months after first symptoms appeared. EEG in earlier disease stage presented slow waves, and PSDs in his later disease stage. [38]. Octapeptide insertions were also mentioned by the CJD Surveillance Committee of Japan, which performed an extensive study on patients with prion disease. This 10-year-long study was established in 1999, and they performed prospective surveillance on human prion diseases. Experts, involved in this study collected data (including imaging, EEG, genetic, biomarker, or neuropathology) from patients. Patients who were suspected of having prion disease were investigated by this study. [39]. Insertion was detected in three patients, who developed disease between 26 and 55 years of age. The disease duration varied from 3–44 months. At least one patient had a positive family history of disease. Two of the three patients presented with periodic sharp wave complexes on EEG. No further details regarding their clinical phenotypes were mentioned [39]. 

The P102L mutation is relatively common among Japanese patients with GSS. The CJD Surveillance Committee has reported mutations in 39 patients. Most patients presented with a familial form of GSS, where the age of onset varied widely. In some, the onset of GSS occurred in their 20s [40,41], while most patients developed GSS in their 50s to 70s [39,42,43,44,45]. All patients displayed motor impairments that included ataxia, walking difficulties, gait dysfunctions, and leg hyperreflexia. Dementia and personality changes were common among affected individuals [40,43]. One family also developed an atypical form of GSS, in which the affected individuals also developed schizophrenia. In this case, cerebral PrP plaques could not be detected by Congo red staining [40]. Interestingly, insomnia also appeared in one family harboring the P102L mutation [41]. Positive cases for 14-3-3 have been reported among patients with P102L; however, in the majority of cases, 14-3-3 was negative or not analyzed [41,45]. GSS associated with P105L mutation has been described in several Japanese patients. The age of onset ranged from to 31 to 57 years, and the disease duration could be prolonged (up to 10 years). Most patients presented with familial or sporadic GSS, but one family was diagnosed with familial parkinsonism [39,46]. The main symptoms are spastic paraparesis [47,48], gait disturbances, and ataxia [49], and tremor [46]. Dementia [47] and memory dysfunction [49] could also be common symptoms. Although spastic paraparesis may occur commonly among patients with P105L, Iwasaki et al. reported a case of GSS involving P105L that did not present with spastic paraparesis [50]. Kubo et al. reported a familial GSS case that presented with atrophy in the frontal and temporal lobes, but not in the cerebellum or occipital lobes. Prp-positive plaques were present in the patient’s sister [51]. Ishizawa et al. [52] described three patients with GSS. Two patients developed dementia with spastic paraparesis and the third developed dementia with psychiatric symptoms without spasticity. PrP plaque deposits were present in all three cases, and phospho-tau also appeared among the plaques. Amyloid plaques were also observed in one patient [52]. One of the most recent cases of P105L developed severe cognitive and gait disturbances and parkinsonism. Swallowing dysfunction was also present in this patient [53]. 

M129V has been suggested as a risk-modifying factor for prion disease. However, one paper [54] described the putative impact of this mutation on multiple system atrophy. The patient harbored the MM homozygous form of M129V and presented with rigidity, speech disturbance, paranoia, and seizures at 60 years of age. CSF was positive for 14-3-3 protein, and EEG revealed slow bilateral sharp complexes. This study revealed that MM and VV genotypes in PRNP codon 129 may increase the risk for multiple system atrophy (MSA) in patients with PD [54]. 

The Y145X stop codon mutation or “amber mutation” has been associated with a unique disease phenotype. In one report [55], the disease started in a female with slow progression in her late 30s. She was initially diagnosed with AD. Disorientation and communication deficits subsequently appeared. The patient died at 59 years of age. Postmortem examination of brain tissue revealed AD-type pathology without spongiform changes. Truncated prion proteins were found in the plaques. The family history of patients with this mutation remained unclear [55]. 

Another nonsense mutation, PRNP Y162X, has been reported in two cases in Japan. The mutation was related to atypical symptoms, such as diarrhea, and refractory esophageal achalasia was present in both cases. In the first case, a mutation presented with diarrhea at 35 years of age. In her 40s, urinary retraction appeared. In her 50s, she developed syncope due to orthostatic hypotension and started to vomit regularly. Significant weight loss was observed during the appearance of the first symptoms. Treatment of the patient resulted in improvement of the esophageal stenosis, but vomiting persisted. Biopsy revealed fine granular prion deposits in the esophageal area [56]. The second patient developed regular diarrhea and esophageal achalasia at 35 years of age. At 53 years of age, visual impairment, such as included tunnel vision or difficulties in seeing in the dark, also appeared. MRI revealed optic nerve atrophy [57]. A frameshift mutation resulted in a 2 bp deletion at residue 178, resulting in a premature STOP codon at residue 203. The patient developed young-onset hereditary prion disease at 26 years of age. Symptoms included cognitive impairment, heart failure, urinary retention, and hypothermia. The disease had a long duration, and the patient died 11 years after the initial symptoms. Postmortem analyses revealed prion deposits in central nervous system and peripheral nerves. The family history was positive, and her mother, maternal grandfather, and younger brother developed similar disease hallmarks. Mutations may result in disease phenotypes owing to missing glycosylphosphatidylinositol (GPI) anchors [58]. The first report of PRNP D178N was observed by the CJD Surveillance Committee in four patients. Three cases were reported with D178N with MM genotype and de novo FFI. The age of onset was 46 to 57 years. PWSC and MRI hyperintensities did not appear in these patients, but in at least one, CSF was positive for 14-3-3. One case of D178N with 129MV genotype was also observed in a patient who developed sporadic prion disease at 79 years of age, and CSF was positive for 14-3-3. However, no abnormalities were observed on MRI or EEG [39]. In 2010, a 54-year-old patient with FFI was reported. His symptoms started with dysphagia and loss of appetite, followed by sleeping dysfunctions (abnormal movement during sleep, insomnia or hypersomnolence, later sleep apnea). Other symptoms also appeared, such as tremor, hyperhidrosis, constipation, and impotence or ataxia. MRI showed mild atrophy, but EEG did not reveal any abnormalities. Histology showed spongiform changes in cingulate gyrus and subiculum, gliosis in thalamus and in inferior olivary nucleus. Immunohistochemistry did not show PrP^Sc^ deposits, but Western blot analysis showed low amount of type 2 PrP^Sc^ and the PrP^Sc^, which had the FFI-type glycosylation pattern. The patient’s mother was diagnosed with rapid progressive dementia at the age of 60. She developed movement issues, such as rigidity and brisk in tendon reflexes, but no ataxia. EEG did not reveal any PSD. Even though she developed CJD-like symptoms, she had similar PrP^Sc^ as the proband patient [59]. An additional case of mutation was related to a family with FFI mimicking DLB. The patient’s mother was diagnosed with CJD, suggesting a positive family history. The proband had weak voice 4 months before hospital admission, at 58-year-of-age. Later, other hallmarks appeared, including visual hallucination, dementia, Parkinsonism (such as ataxia, tremor, and bradykinesia), weight loss, or impaired speech. Sleep disturbances appeared too, such as reduced sleep time, loss of rapid eye movement during REM phase. The patient was suspected of having DLB because he developed hallucinations and Parkinsonism, but symptoms progressed more rapidly than in typical DLB. The ^123^I-ioflupane SPECT analysis was performed for dopamine transporter binding analysis, which showed rapidly declining functions in the presynaptic striatal dopaminergic nerve terminal. Swallowing dysfunctions and aspiration pneumonia appeared at 7 months after disease onset, and the patient died 2 months later due to pneumonia. CSF samples were negative for 14-3-3 [60]. V180I commonly occurs in Japanese patients with prion disease (>100 cases). Most patients develop CJD [39], but other phenotypes have also been reported, such as AD-type pathology CJD [61], DLB [62], or PD-like CJD [63]. V180I cases were mostly associated with late disease onset (>70 years of age), but some patients started disease phenotypes prior to 65 years of age [64,65]. The majority of cases with V180I in Japan featured a negative family history, but the familial form of V180I. However, a few cases with positive family history were also observed [39]. CSF 14-3-3 signal has been reported as positive [63] and negative signals [65,66]. Brain hallmarks may vary among patients. Affected areas can be the left cerebral cortex, bilateral cerebral cortex [67,68,69], or basal ganglia [69]. One CJD patient presented with spongiform changes and non-confluent (VaSNoC) vacuoles [70]. However, large vacuoles [71] and acute cerebral infarction [61] also appeared in CJD patients with V180I. In patients with AD-type pathology, senile plaques, neurofibrillary tangles, and spongiosis have been observed in the brain [61]. E200K appeared in several familial or sporadic CJD cases, where the disease occurred between the ages of 31 and 78 years. Symptoms included progressive dementia, myeloid jerks, insomnia, psychomotor excitement, and visual disturbance. PSD was present on EEG. The majority of patients were positive for 14-3-3. Astrogliosis and spongiform changes were also prominent in the patients. It is possible that these patients had a “founder effect” of mutation, as its frequency in the Fuji area was significantly higher, compared to other regions in Japan. The same study revealed that Japanese patients with E200K mutations presented with similar clinical symptoms and neuropathological hallmarks, compared to E200K-CJD cases from other ethnical groups (such as Lybian Jews or Europeans) [39,72]. V203I was initially observed in 2010 in two sporadic CJD cases in the 70s. Disease progression was rapid, occurring within 6 months, and both patients had PWSCs on EEG and hyperintensities on MRI. CSF of one patient tested positive for 14-3-3. No detailed description was given on their disease symptoms and clinical progression [39]. A homozygous V203I mutation was also discovered in a homozygous in a 73-year-old CJD patient. The patient developed rapidly progressive disease, progressive gait disturbance, and cognitive dysfunction. She became bedridden 3 months after onset, and developed akinetic mutism with myoclonus after 4 months. Disease duration was relatively long (24 months). MRI revealed abnormalities in different brain areas, such as right basal ganglia and the right frontal, parietal, and occipital cortices. EEG presented slow waves in earlier disease stage, and later PSDs. CSF 14-3-3 and phospho-tau levels were elevated. However, no differences were found between the phenotypes and disease courses of the current homozygous case and previously described heterozygous forms of V203I [73]. The R208H mutation was observed in one sporadic CJD patient who developed rapid progressive CJD at 74 years of age and CSF was positive for 14-3-3 [39]. The I210V mutation appeared in a CJD case with unknown family history at 58 years of age. The patient initially had a cerebral infarction, followed by memory, gait disturbance, and myoclonus. At the age of 69 other symptoms appeared, such as progressive memory decline, gait disturbance, and personality changes. After 2 months, akinetic mutism and myoclonus appeared. EEG showed PSED in the late disease stage. Brain imaging revealed moderate atrophy, but no fresh hemorrhage or infarction. CSF samples were negative for 14-3-3 [74]. M232R mutation commonly occurs in Japanese patients, and most patients had either sporadic CJD [39,75,76] or atypical CJD [77,78,79], with rapidly progressive dementia, behavioral issues, or sleep disturbances. Additional non-CJD cases have also been reported among patients with M232R, such as DLB [80,81,82] or slow progressive ataxia [78]. The age of onset ranged from the late 50s to 70s. One patient presented with a homozygous mutation, representing the first case of homozygous M232R. The patient had elevated CSF 14-3-3 and tau levels. The homozygous M232R was associated with earlier disease onset and faster disease progression than the heterozygous form of mutation. It may be possible that homozygous mutation may accelerate the disease progression [78]. Two CJD patients were compound heterozygous for V180I and M232R mutations. The patients developed diseases at 74 years of age. One patient presented with cognitive dysfunction, dysarthria, and gait disturbance, but the disease progressed slowly. Elevated levels of tau and 14-3-3 have been found in the CSF. The present case may be related to an atypical form of CJD. The other patient tested negative for CSF 14-3-3, but presented with PWSCs on EEG [39,83]. The findings are summarized in Table 2 and Figure 2b.

**Table 2 ijms-24-00625-t002:** Japanese cases of prion mutations (AOO means age of onset, hm means homozygous).

Mutation	Disease	AOO (Years)	Family History	Phenotype	14-3-3	EEG	Imaging	Remarks	Reference
144 bp insertion	Familial prion disease	late 20s–30s	+	Dementia, ataxia, extrapyramidal symptoms	NA	PSD or slow waves	MRI: diffuse cerebral and cerebellar atrophy	PrP + plaques	[37]
4-OPRI	RPDS	56	−	Rapidprogressive ataxia, myoclonus	NA	slow waves	MRI: moderate cerebellaratrophy	NA	[38]
OPRI	CJD	26–55	1/3 +	3 cases of CJD	1/3 −	PWSC in 2 patients	One had Hyperintensities on MRI	NA	[39]
P102L	GSS	34	+	Dementia or ataxia or both	−	Minor abnormalities	MRI: moderate in cortexatrophy	PrP + plaques	[40]
	GSS	74	+	Ataxia in limbs and speech, paresthesia and areflexia	−	Normal	Normal	NA	[45]
	GSS	38	+	Unsteady gait, dysarthria, insomnia, mental issues	+	Diffuse d andt waves	MRI: mild vermin atrophy	Elevated CSF-Tau	[41]
	GSS	22–75	74% +	39 patients with familial or sporadic GSS	15% +	19% PSWC	39% hyperintensities on MRI	NA	[39]
	GSS	55 and 66	+ or NA	2 cases of language impairment, dysphagia	NA	NA	MRI and SPECT: Thalamus abnormalities	Elevated CSF-Tau	[44]
	GSS	59–74	+	5 family members: ataxia, gait disturbance, dysesthesia, dementia	NA	Normal	SPECT and PET: mosaic-like pattern of blood flow and glucose metabolism	NA	[43]
	GSS	56	+	Worsening dizziness and walking instability, ataxia	NA	BIPDs	MRI: mild cerebellar atrophy	NA	[42]
P105L	GSS	38–47	+	5 cases of spastic paraparesis, mild dementia	NA	normal	NA	amyloid plaques	[47]
	GSS	42	−	Spastic paraparesis, ataxia, memory dysfunctions, dysarthria, m apraxia	NA	a waves, low t and d activities	CT and MRI: cortical atrophy	PrP + plaques	[49]
	GSS	46	+	Weakness in legs, spastic gait	NA	NA	MRI: atrophy of frontal and temporal lobes,	Prp + plaques	[51]
	GSS	57	+	Dementia, gait disturbance, spastic disturbance	NA	NA	CT and MRI: atrophy of the frontal lobes	Amyloid plaques and NFTs	[48]
	GSS	36	+	Memory decline, dysarthria, extrapyramidal signs, ataxia, no spasticity	−	NA	MRI: atrophy in cerebral white matter, SPECT: reduced blood flow in different brain areas	Elevated CSF-Tau	[50]
	GSS	31–51	+ or −	5 cases of familial or sporadic GSS	−	PSWC were rare	MRI: low rate of hyperintensities	NA	[39]
	GSS, PD	40–53	+	3 families, dementia, gait disturbance	−	Normal	MRI: mild diffuse atrophy of the cerebral cortex	NA	[46]
	GSS	35–45	+	3 members of a family, dementia and spastic paraparesis	−	Normal	MRI: mild frontotemporal atrophy	PrP plaques with p-Tau and amyloid beta	[52]
	Parkinsonism	54	+	Parkinsonism, swallowing issues	NA	NA	NA	NA	[53]
M129V	MSA	60	NA	Rigidity, speech disturbance, paranoia, seizures	Positive	bilateral sharp complexes	DWI MRI: increased signal in cortex and putamen	NA	[54]
	NA	NA	NA	Na	NA	NA	Risk modifier factor	NA	[39]
Y145X	GSS, dementia	45	NA	AD-like phenotypes	NA	NA	NA	AD pathology, Prp+ plaques	[55]
Y162X	Optic nerve atrophy	56	NA	Gastrointestinal symptoms, refractory esophageal achalasia, visual issues	NA	NA	MRI: optic nerve atrophy	NA	[57]
	Refractory Esophageal Achalasia	55	+	Diarrhea, urinary retraction	NA	NA	NA	Prp deposits is esophageal area	[56]
D178fs	Hereditary Prion Disease	37	+	Cognitive impairment, heart failure, urinary retention, hypothermia	NA	NA	MRI: normal	PrP deposits in brain	[58]
D178N	FFI-129M	54–60	+	2 cases of dysphagia, sleep issues, memory decline, ataxia, visual hallucinations	NA	Normal	MRI: mild atrophy; SPECT: cerebral cortex and thalamus, hypoperfusion and hypometabolism	Histology: spongiform changes	[59]
	CJD-129V	74	−	Sporadic prion disease	+	No PSWC	MRI: no hyperintensities	NA	[39]
	FFI-129M	46–57	−	3 cases of de novo FFI	1/3 +	No PSWC	MRI: no hyperintensities	NA	[39]
	FFI, DLB-like	58	NA	Insomnia, ataxia, tremor, bradykinesia	+	slow waves, no PSD	MRI: diffuse cortical atrophy	NA	[60]
V180I	CJD	58–81	−	9 cases if slow progressive CJD, myoclonic jerks, akinetic mutism	3/9 +	No PSWC	MRI: cortical lesions, swellings	NA	[88]
	CJD	67–74	−	3 cases of memory loss, language impairment, myoclonus was missing in a patient	-	No PSD	MRI: cortical ribbons, cortical or basal ganglia lesions	Elevated CSF-Tau	[85]
	CJD	44–93	−	98 cases of sporadic CJD, slow progression	78% +	PSWC was rare	MRI: hyperintensities	NA	[39]
	CJD/AD	77	−	Dementia, gait disturbances, ataxia	+	No PSD	MRI: diffuse cerebral cortical atrophy, cortical lesions	Senile plaques and NFTs in brain	[61]
	CJD	75	−	Aphasia, amnesia and unsteadiness of gait	NA	Slow wave, no PSD	SPECT: hypoperfusion in left parietal and frontal lobes, bilateral basal ganglia, reduced blood flow	CSF: increased prion levels	[68]
	CJD	76–82	−	3 cases of dementia, parkinsonism, behavioral dysfunctions	2/3 +	Slowing but no PSWC		MRI: gyriform hyperintensity or swelling	[85]
	CJD	70s	+	Memory dysfunctions, bradykinesia but no myoclonus	+	Generalized slow basic rhythm	MRI: increased signal intensity in bilateral frontal, temporal, parietal cerebral cortex	Elevated CSF-Tau	[68]
	CJD	57	NA	Slow progressive CJD, cognitive impairment	NA	NA	MRI: high-intensity areas in the cerebral cortex and basal ganglia, but thalamus, brainstem and cerebellum were preserved	Long disease duration, over 10 years	[64]
	CJD	78	−	Weakness and gait disturbance, later cognitive impairment	− Then +	Slow wave, no PSD	MRI: asymmetric cortical high intensity	CSF: elevated Tau	[86]
	CJD	69–78	−	3 cases of dementia, parkinsonism, no cerebellar signs or visual dysfunction	+	No PSWC	SPECT: Preserved cerebral blood flow in different brain areas		[84]
	CJD	78	−	Slow progressive disorientation, memory dysfunctions, myoclonus, pathological laughing	+	Low basic pattern but no PSWCs.	MRI: extensive hyperintensity of cerebral cortex	Spongiformvarious-sized and non-confluent (VaSNoC) vacuoles	[70]
	CJD	73–87	−	6 cases of dementia, tremor, behavioral changes	+	PSWC and slowing	MRI swelling, hyperintensities	Larger vacuolar sizes	[71]
	CJD	87	−	Facial mimicry, cognitive dysfunction, gait disturbance	+	Diffuse slow basic pattern	MRI: cerebral hyperintensity and swelling	NA	[87]
	DLB	75	−	Cognitive dysfunction, tremor, hyposmia	-	Slowing without PSWCs	MRI: hyperintensities in cerebral cortices	NA	[62]
	CJD	64	−	Headache, rapid progressive dementia, gait disturbance, no myoclonus	+	No PSWC	MRI: hyperintensity in right parietal cortex	acute cerebral infarction	[65]
	CJD	74	NA	Cognitive decline, rapid progressive	−	No PSD	MRI: cortical swelling	NA	[66]
	CJD, mimicking PD	79	−	Gait disturbance, cognitive dysfunctions, bradykinesia, rigidity and tremor	+	diffuse slow basic pattern, no PSD	MRI: hyperintensity in bilateral cerebral cortices	Elevated CSF-Tau	[63]
	CJD	84	−	Initially misdiagnosed to DLB and person delusional misidentification	+	Normal	MRI: showed cortical hyperintensities	CJD	[67]
E200K	CJD	50–78	+ or −	6 cases: dementia, myeloid jerks, insomnia, psychomotor issues, visual disturbance	NA	PSD	NA	Spongiform encephalopathy	[72]
	CJD	44–78	+ or −	37 cases of familial or sporadic CJD	+	PSWC	MRI: hyperintensities	NA	[39]
V203I	CJD	73	−	2 cases of sporadic CJD	1/2 +		MRI: hyperintensities	NA	[39]
	CJD-hm	73	−	gait disturbance, cognitive decline akinetic mutism	+	Diffuse slowing waves	MRI: increased signal intensity in different brain areas	Elevated CSF-Tau	[73]
R208H	CJD	74	−	Slow progressive CJD	+	PSWC	MRI: hyperintensities	NA	[39]
V210I	CJD	58	NA	Cerebral infraction, memory-and gait disturbance, myoclonus	−	PSD	MRI and CT: moderate brain atrophy	NA	[74]
E219K	NA	NA	NA	NA	NA	NA	Risk modifier, possible protective against sCJD	NA	[39]
M232R	CJD	50s–60s	−	20 cases of rapid or slow progressive CJD, akinetic mutism, myoclonus	+	PSWC was rare	MRI: high intensity lesions in different brain areas	NA	[71]
	DLB	55	−	Progressive dementia, gait dysfunctions	NA	No PSD	MRI: slight atrophy, SPECT: hypoperfusion in the bilateral occipital cortices	Lewy bodies, no spongiform changes	[76]
	CJD	69	−	Memory disturbance, apraxia, myoclonus	+	Unstable 9-Hz alpha waves	MRI: high intensity in cortical ribbon, bilateral medial thalami	NA	[72]
	CJD	15–81	−	33 cases of sporadic CJD	+	common PWSC	MRI: hyperintensity	NA	[39]
	Ataxia	57	−	Dysarthria, gait, slow progressive ataxia	−	Normal	MRI: normal	NA	[78]
	CJD	60	−	Behavioral issues, gait disturbance bradykinesia, slow progressive dementia	NA	Atypical PSWC	MRI: brain atrophy with ventricular dilatation, DWI: high signal in cerebral cortices	Protein resistant PrP type 1+2 were present	[73]
	CJD-hm	50	−	Rapid progressive dementia, gait disturbance	+	PSWC	MRI: hyperintensity in bilateral cerebral cortex and striatum	Elevated CSF-Tau	[74]
	DLB/CJD	77	−	Rapid progressive dementia, Parkinsonism	−	PSWC	high signal in right temporal lobe.	Elevated CSF-Tau	[77]
	CJD	54	−	Slowly progressive dementia and sleep disturbance	NA	Diffuse slowing, no PSD	MRI: bilateral diffuse high signal in frontal, parietal, and temporal cortices, striatum, and thalamus	Amyloid plaques in cerebellum	[75]
V180I+ M232R	CJD	74	−	Sporadic CJD	−	PSWC	MRI: no hyperintensities	NA	[39]
	CJD	74	NA	Dysarthria, gait disturbance, cognitive impairment, slow disease progression	+	Atypical PSD	MRI: high signal in bilateral cortex	Elevated CSF-Tau	[78]

## 4. Prion Mutations in China

Several studies are available on Chinese patients with prion disease and mutations. However, it may be possible that there are several additional patients who remained undiagnosed for prion disease, especially from undeveloped areas of the country. Mutations in PRNP in China were significantly different in comparison to Korea or Japan. Several unique prion mutations appeared in Chinese individuals, such as T188K or E196A. Furthermore, V180I and M232R were very rare in China. These differences could be explained by geographical reasons [89,90,91,92,93,94,95,96,97,98,99,100,101,102,103,104,105,106,107,108,109,110,111,112,113,114,115,116,117,118,119,120,121,122,123,124,125,126,127]. Additional rare prion mutations have been described in CJD, FFI, GSS, AD, and PD patients (Table 3, Figure 1c). The S17G mutation appeared in a case of late onset AD (LOAD). The affected patient developed the disease at 70 years of age, and disease progression was slow. Symptoms included memory dysfunction, language impairment, and personality changes. Motor function remained normal. MRI showed diffuse cortical atrophy in the cortex, enlargement of the cerebral ventricles, and the cistern in most brain areas, such as in the frontotemporal lobe or hippocampus. It may be unclear whether the PRNP S17G could impact the disease phenotypes [89]. Single octapeptide deletion may not be associated with prion diseases, but may be a risk factor for gastric cancer in Chinese patients. Octapeptide deletions were found to be more frequent among cancer patients than in normal individuals. The deletion may result in a higher degree of proliferation in cancer cells, but may not affect cell apoptosis, adhesion, or invasion [86]. Another report described a single octapeptide deletion 58-year-old PD patient and suggested that the deletion may be a risk factor for PD [90]. Symptoms started with tremor of upper left limb and dizziness. After 6 months, she had more symptoms, such as slowness, sleep impairment, and was hospitalized with multiple conditions, including PD, sleep disorder, hypertension, and coronary heart disease. Later, her memory was also deteriorating. The authors reported that CSF 14-3-3 was negative, and EEG revealed no PWSC, but mild abnormalities. MRI revealed a low degree of white matter myelination [91]. 

Two cases of OPRI were observed in the Chinese Surveillance Program. One patient harbored a single octapeptide insertion associated with prion disease at 58 years of age. The other patient harbored a seven-octapeptide repeat insertion that was associated with prion disease at 42 years of age. Both patients developed dementia, myoclonus, and pyramidal-extrapyramidal dysfunctions, but the seven-octapeptide repeat carrier also presented with visual and cerebellar dysfunctions [92]. The S97N mutation was found in a probable AD patient who developed disease phenotypes at 70 years of age. She experienced memory issues, and she lost the ability to perform her daily activities. A year after, her cognition worsened, and she developed visual impairment. Corticospinal syndrome has also been observed in patients with bilateral deep tendon hyperreflexes. MRI revealed mild cerebral atrophy [93]. 

The P102L mutation has been reported in several Chinese GSS cases, but it may not be as common mutation in China as in Japan. The age of onset ranged from 34 to 67 years, when patients developed motor impairments, ataxia, or dementia. 14-3-3 positivity and PSW on EEG was relatively rare [94,95,96]. In 2017, Li et al. [94] reported five GSS cases with the P102L mutation. The patients developed the disease between 37 and 59 years of age. All cases started with walking difficulties, and progressive ataxia was one of the main hallmarks. Phenotypic heterogeneity was observed among affected family members. Although cognitive decline was common among them (four of the five patients), only two of them presented with early symptoms. Two individuals presented with cognitive dysfunction in the later disease stage (2 years after initial symptoms). The study findings suggest that phenotypic diversity may be related to unidentified genetic or environmental factors [94]. In 2017, Long et al. [95] reported a case of familial GSS. The female proband developed unstable gait and dysarthria at 44 years of age. In addition, she experienced choking after drinking, and her speech slurred. The symptoms worsened after cervical vertebral surgery for cervical disc hernia. However, the patient did not experience any memory loss or neuropsychiatric symptoms. She showed dysarthria, nystagmus (both horizontal and vertical), wide base, and unsteady gait. Brain MRI revealed mild diffuse atrophy and caval vergae. The patient’s spinal cord was also damaged by herniation [95]. A 2019 study analyzed 12 GSS patients with the P102L mutation that were included in the Chinese National Surveillance Network by the Chinese CDC [92,96]. The majority of these patients displayed movement impairment as an early symptom. This could be followed by various symptoms that included dementia or mental issues. Rapid progressive dementia was more common (seven of the 12 patients), compared to slow progressive dementia (two patients). The 14-3-3 marker was present in five of 11 cases, while PSWC was present in 25% of the cases. Several patients display sporadic CJD-like (confusion, depression, memory issues) symptoms. MRI revealed sporadic CJD-like abnormalities, such as a high signal in the caudate or putamen. The findings suggest that sporadic CJD-like symptoms are common among Chinese patients with GSS. In addition, E219K has been suggested as a potential risk modifier in cases harboring the P102L mutation. Further studies are needed to verify these findings [96,97]. The most recently reported P102L patient was initially diagnosed with cervical spondylitis myelopathy at the age of 49 years. Patient experienced unsteady walk a year before hospital admission. These symptoms did not improve after cervical discectomy surgery. At the second visit (2 years after the first visit), other symptoms appeared, such as slowness of pharyngeal reflex, abnormal speech, supination orthostatis, and blood pressure changes. Cognitive functions also dropped. The third visit (a year after), additional dysfunctions appeared, such as anxiety, personality changes, sleep issues, and reduced reflex function. Even though MRI was normal, DWI revealed serious brain atrophy. The patient was negative for spinocerebellar ataxia-related repeat expansions. This diagnosis was revised to GSS after the discovery of P102L mutation [98]. P105L may be a rare mutation among Chinese patients, as only one case of GSS mutation has been reported. The patient developed the disease at a young age (11 years), but the disease duration was long. No further details regarding the patient’s disease phenotypes and disease course have been mentioned [99]. 

G114V has been reported in familial prion disease, which occurs in patients in their 40s or 50s. A large family with 49 members (including spouses) was analyzed with this mutation. Proband developed progressive dementia at the age of 45, and also experienced tiredness, lethargy, and sleep issues. She also had motor issues, such as myoclonus, Babinski sings, or hyperreflexia. MRI revealed bilateral atrophy in different brain areas. In this family, other members with similar mutations were found to be carriers of G114V, including the two siblings of the proband. Also, one relative (son of her first cousin) experienced progressive memory impairment and ataxia at the age of 32 [100]. Mutation also appeared in two familial cases of CJD in 2015. G114V was screened in the current preclinical carrier family members of one of the patients with CJD [101,102]. Diffusion tensor imaging was used to analyze fractional anisotropy and mean diffusivity (MD) in the white matter of preclinical individuals. Preclinical individuals show white matter alterations, such as increased MD in different brain areas [101]. FDG-PET of asymptomatic carriers revealed reduced metabolism in different brain areas, including the thalamus, postcentral, left fusiform, left superior temporal, left lingual, left superior parietal, and left Heschl’s gyrus [102]. The V allele of M129V has been examined in patients with mesial temporal lobe epilepsy (MTLE). In 2007, a similar study reported that M129V may be a possible risk modifier for MTLE in the Italian population. However, the Chinese study could not confirm the association [103]. 

R148H was detected in a patient with sporadic CJD at 68 years of age. Symptoms include dementia, pyramidal or extrapyramidal dysfunction, and akinetic mutism, but no further details were published on the disease progression [92,95]. D178N is very common among Chinese patients and is related to both FFI and CJD. Interestingly, CJD cases are also associated with the M/M genotype at codon 129 [92,93,104,105]. Chen et al. (2015) published a case of with D178N-129MM. The patient presented rapid memory loss at the age of 45. She was also disoriented and had speech impairment, but her sleeping remained normal. EEG revealed slow brain activity, and MRI revealed ribbon like high signal in multiple brain areas. She was diagnosed initially with sporadic CJD. Three months after first examination, insomnia and motor impairment (myoclonus) appeared in her. Akinetic mutism and incontinence appeared 8 months after first hospital visit, but insomnia disappeared. MRI in later disease stages revealed diffuse cerebral atrophy. Disease duration was 16 months [104]. Huang et al. (2020) also described a familial CJD case with D178N-129MM. The proband patient was a 58-years-old female, whose initial hallmarks were motor impairments (bradykinesia, hypomimia, stiffness), swollen wrist, weight loss, but no insomnia or any sleep impairment. Cognitive functions were also impaired. A year after first hospital visit, other symptoms appeared (arm spasm, dysphonia), and the patient became bedridden. Family history was positive [105]. Lu et al. described a patient who displayed white matter hyperintensities and CADASIL co-pathology. The patient was a female patient in her late 50s who experienced hypertension several years before insomnia appeared. Motor impairment (tremor), and personality changes also appeared. The patient also transient global amnesia (TGA) a year after insomnia, followed by autonomic dysfunctions. Brain biopsy revealed spongiform encephalopathy, astrogliosis, arteriosclerosis, and PrP staining was also positive. The D178N mutation co-existed with a pathogenic NOTCH3 variant (R544C), but the reason of co-existence remained unclear [106]. One family with FFI showed FTD-like symptoms that included anxiety about dementia [107]. In 2021, Yukang et al. reported a case that also presented with depression in addition to FFI [108]. 

The V180I mutation has been relatively rare in China compared to Kora or Japan. The mutation has been described in only two patients developed CJD in their 70s. The initial symptom was bradykinesia. Rapid progressive memory issues also appeared. Myoclonic jerks and muscle tone appeared 3 months after first symptoms. Six months after disease onset, she had akinetic mutism. Disease duration was longer than one year. EEG results were normal, and one patient was negative for 14-3-3 [109]. The second case presented with PSWC on EEG, but was also negative for the 14-3-3 signal, but no details were mentioned on the clinical symptoms and disease progression [92]. 

The T183A mutation was reported in a sporadic CJD patient who developed dementia at 42 years of age. No details were described on disease progression. CSF 14-3-3 was negative, and Tau in CSF was elevated. RT-Quic of the patient was positive [99]. The T188K mutation has been implicated as a common causative factor of CJD among Chinese patients. [92,110]. Wang et al. described 3 sporadic CJD cases with T188K. The first patient started to experience depression and insomnia in his 60s. Short-term memory was also impaired. In a year, these symptoms worsened, and other hallmarks appeared, including mutism, hypologia, somnolence, visual issues, or motor issues. The second case was a female patient in her 60s who developed initially memory and language issues and upper limb myoclonus. Symptoms worsened, and additional issues also appeared, such as jerky myoclonus or hallucinations. The patient died 5 months after first symptoms due to pneumonia. The third patient was in his 70s, who developed hyposthenia initially. Memory also declined. Several types of motor impairment also appeared during disease progression, including ataxia, muscle weakness and myoclonus. The patient died 10 months after first symptoms [111]. Chen et al. (2013) reported eight CJD cases of T188K, but only one of them had positive family history of disease. Disease occurred between 39–76 years of age. Variable initial symptoms were reported. The majority of patients (63%) had rapid progressive dementia as initial symptoms, followed by walking instability (50%). In later disease stages, all patients developed dementia and pyramidal or extrapyramidal issues. Myoclonus and cerebellar dysfunctions were also commonly occurred, but akinetic mutism was also reported [113]. Shi et al. (2017) analyzed 30 patients with T188K. The majority of patients did not have any family history of disease. Progressive dementia was relatively common (~66%) initial symptom among them. Additional initial hallmarks were cerebellar issues and unstable walking (~40%), mental dysfunctions (~33%), and extrapyramidal symptoms (30%). During disease progression, almost all patients experienced progressive dementia, but motor impairment were also common, including pyramidal and extrapyramidal issues or myoclonus. Visual disturbances and akinetic mutism also occurred in the late disease stages [112]. Most patients have been positive for 14-3-3; however, negative cases have been described [92,110]. EEG has revealed abnormalities in patients with T188K, such as PWSC [110,111,112] and PSG [109]. One homozygous case of T188K has been described, but the phenotype and age of onset were similar to those of the heterozygous form of the mutation [110]. 

The E196K mutation was identified in a 71-year-old patient with an atypical CJD. The patient displayed initially rapid progressive dysfunctions in speech. Additional symptoms also occurred: hypomnesia listlessness or dysphagia. Memory, cognition, and physical movements were also impaired. In the final disease stages, akinetic mutism, coma, and cachexia appeared, and the patient died 6 months after first symptoms. EEG was normal as was CSF 14-3-3 in the CSF [114]. 

E196A may be a relatively common mutation among Chinese CJD patients, as at least nine cases harboring this mutation have been described [115,116,117,118]. The first patient was described in 2014, in his 70s, and initially developed intellectual dysfunctions, and was suggested to have intermittent mental and behavioral disorders (including hypochondria). Speech dysfunctions also appeared. Six months before being admitted to medical facility, his motor functions, sleeping, and diet remained initially normal. Myoclonic jerks appeared a week after admission. In a month, his conditions became worse, he developed other dysfunctions, such as insomnia, urinary incontinence, aphasia, ataxia. The patient died in 2 months after the first symptoms [115]. Shi et al. (2016) reported two additional, probably unrelated patients with E196A with sporadic CJD. The first patient was a 54-years-old female with intellectual decline and unsteady walking. These symptoms worsened fast. Patient became unconscious and started to have reflex issues. Disease length was 22 months. The other patient experienced speech issues and hand weakness initially. She also had dysarthria, emotional instability upon hospitalization. Later, motor impairment became more serious (inability to walk steadily, eye opening issues, later myoclonus). Akinetic mutism appeared after 6 months. [116]. Dai et al. found E196A in a 56-year-old patient, who experienced progressive movement disorder with different motor issues (limb weakness, tremor, gait issues). Initially, no cognitive impairment appeared, but during hospitalization, her cognition started to decline. After 2 months, the symptoms started to became worse, myoclonic jerks, psychotic issues, and akinetic mutism appeared. No neuropathological data are available. Patient was also suspected to have thyroid cancer [118]. Wu et al. reported one additional patient with probable sporadic CJD, who experienced memory issues, visual symptoms (metamorphopsia, difficulties of seeing colors), and disorientation 2 months before hospital admission. Later, cortical blindness and motor issues (tremor, increased muscle tone) also appeared, followed by hallucinations. The patient’s mother and daughter also carried the same mutation, but were asymptomatic. It is unclear, whether the patient’s mother carried any possible neuroprotective genetic or environmental factors [117]. All patients were tested positive for 14-3-3. Intellectual and behavioral dysfunction can occur in patients harboring the E196A mutation [115,116,117], and motor issues have also been described [114] EEG examinations revealed PWSC [116,117] or PSD [115]; one case featured no abnormalities [118]. 

The F189V mutation appeared at 56 years of age in a Chinese patient with a probable case of AD. The patient developed memory loss and myoclonous hallucinations. MRI and EEG showed moderate and diffuse encephalic atrophy, respectively [93]. 

The E200K mutation may be common in Chinese patients with CJD. The first patient presented with memory and motor disturbances, and 14-3-3 positivity in the CSF. The EEG pattern was atypical in terms of slowing of background activity and periodicity, with triphasic sharp waves [119]. The second case of CJD with E200K mimicked FFI. Patient had initially sleep disorder followed by dizziness 12 months before hospital visit. Initial diagnosis was generalized anxiety disorder (GAD), but his symptoms were not improved after medication. Two months later, motor impairment appeared (leg muscle atrophy, ataxia), and later he could not balance properly. Memory issues, dreaminess, and sleepwalking also appeared. Symptoms worsened with disease progression; patient died 18 months after first disease signs. EEG, MRI, and 14-3-3 CSF findings were normal, but diagnosis of CJD was made after genetic analysis and discovery of E200K mutation [120]. In 2019, 30 Chinese patients with the E200K mutation were analyzed by the China CJD Surveillance Center [121]. The age of onset was 42–71 years. Initial symptoms included progressive dementia and/or recognition issues. Cerebellar symptoms and sleep disturbances also occurred. During disease progression, different dysfunctions that could occur included pyramidal and extrapyramidal symptoms, myoclonus, visual dysfunction, and akinetic mutism. The majority of patients were positive for the 14-3-3 marker, and half showed PWSC on EEG. Disease duration ranged from to 2–26 months. A putative risk modifier of CYP4X1 (rs9793471) could be present [121]. 

The E200G and mutation was found in a 63-year-old patient with sporadic CJD. The mutation was associated with pyramidal or extrapyramidal dysfunction [99]. The E200A mutation was reported in a CJD patient without any family history who developed CJD at 60 years of age. No changes in MRI were detected, and the phenotypes included dementia, mental issues, myoclonus, and pyramidal signs [92]. The V203I mutation was present in one late-onset CJD patient (80 years of age), who had initially memory and language impairment, dizziness, blurred vision, and ataxia. Symptoms worsened in a week, and he also experienced myoclonic jerks and lost the ability to walk by himself. A week after the memory issues worsened, tremor and myoclonus appeared more regularly. By the final disease stage, he became bedridden with akinetic mutism. The patient tested positive for 14-3-3 [122]. The R208C mutation was described in a patient with probable late-onset slow progressive dementia who presented with hallucinations and cognitive dysfunctions. Short-term memory, calculation, and comprehension were impaired, but no pyramidal, extrapyramidal, or cerebellar dysfunction was reported. MRI revealed moderate encephalic atrophy [93]. Two patients with CJD who were likely unrelated to one another harbored the PRNP R208H mutation. Both patients presented with dementia, myoclonus, and visual and cerebellar disturbances. One patient was positive for the 14-3-3 signal and PSWC on EEG. No MRI data were available for either patient [92]. Two publications are available on Chinese V210I mutation cases in CJD patients [123,124]. One patient developed paraencephalitic CJD at 48 years of age and presented with motor clumsiness, hand dystonia, and cognitive dysfunction one month before hospital admission. She also experienced myoclonic jerks and deep tendon reflexes. Two months later, her symptoms became worse, and she became bedridden with global aphasia. MRI revealed increased signal intensity in the basal ganglia and thalamus. Even though her siblings did not have any neurodegenerative phenotype, two of her children (asymptomatic at the time of study) carried the mutation. [123]. The second case reported three sporadic CJD patients with an age of onset of 59–69 years [124]. All patients experienced rapid disease progression. The first patient presented dizziness, followed by mental dysfunctions (memory loss, speech issues) in his late 60s. Motor impairment also appeared and he died 75 days after first dysfunctions appeared. The second one was a 64 years old man who presented blurred vision, hallucinations, and sleep issues. Ten days later, he experienced other dysfunctions, such as mental impairment, apathy, speech loss, drowsiness, and dysphagia. The patient was still alive in 10 months after disease onset, but had akinetic mutism. The third one was a 59-years-old male patient, who had initially dizziness and body shaking for no reason. Later, he developed instability during walking, speech issues. In three months, he had akinetic mutism and died after first symptoms. EEG showed typical or atypical PWSCs, and CSF was positive for the 14-3-3 protein [124]. M232R has been described but may not be a common variant in Chinese patients. One case of M232R with genetic prion disease has been reported. The patient developed symptoms at 44 years of age. Behavioral changes, cognitive dysfunction, and ataxia were evident. Two years after, his cognition became worse, and his walk became unsteady. Pyramidal signs and mild cerebellar signs were reported too. MRI revealed cerebral and cerebellar atrophy [93]. The findings were summarized in Table 3 and Figure 2c.

**Table 3 ijms-24-00625-t003:** Prion mutations found in Chinese patients.

Mutation	Disease	AOO (Years)	Family History	Phenotype	14-3-3	EEG	Imaging	Remarks	Reference
S17G	AD	70	−	Memory dysfunctions, personality changes	NA	NA	MRI: diffuse cortical atrophy, enlargement of the cerebral ventricle	NA	[89]
octapeptide deletion	Gastric cancer	NA	NA	NA	NA	NA	NA	Higher frequency in cancer patients	[90]
	PD	58	−	Parkinsonism	−	Mild abnormality	MRI: mild write matter demyelination.	NA	[91]
OPRI	CJD	30s–60s	+ or −	2 cases CJD, dementia, myoclonus	1/2 +	PSWC	MRI: positive or negative for hyperintensities	NA	[92]
N97S	AD	74	−	Cognitive –and memory impairment, hallucinations	NA	NA	MRI: mild cerebral atrophy	NA	[93]
P102L	GSS	40s–50s	+ or −	3 cases of GSS, dementia, myoclonus	2/3 −	2/3 PSWC	MRI: 1 had hyperintensities	NA	[92]
	GSS	45–58	+	5 cases of motor impairment, ataxia, dementia, personality changes	NA	Normal	MRI: cerebral atrophy	NA	[93]
	GSS	40s	+	7 cases of cerebellar ataxia	-	Normal	MRI: cavum vergae, mild diffuse brain atrophy	NA	[94]
	GSS	34–67	+	12 cases of movement symptoms, dementia	5/12 +	PSWC rare	MRI: high signal intensities in the caudate/putamen DWI: symmetrical or dissymmetrical cortical ribbon syndrome	NA	[92,97]
	GSS/	49	NA	Gait instability, personality changes, sleep disturbances, cervical spondylitis myelopathy	NA	NA	MRI: brain atrophy	NA	[98]
P105L	GSS	11	NA	NA	NA	NA	NA	Long disease duration	[99]
G114V	CJD	20s–60s	−	3 cases of GSS, dementia, myoclonus	−	Normal	MRI: normal	NA	[92]
	CJD and preclinical CJD	47	+	progressive dementia, tiredness, lethargy, sleep disturbances	NA	Slow waves	MRI: bilateral atrophy of different brain areas	NA	[100]
		50s-60s		7 unaffected 10 affected members	NA	No sign on preclinical CJD	FDG-PET: Hypo-metabolism of parietal and temporal lobe in preclinical patients	DTI: White Matter Integrity Involvement in pre-clinical stage	[101,102]
M129V	MTLE	NA	NA	NA	NA	NA	NA	Not related to epilepsy	[103]
R148H	CJD	60s	+	Dementia, myoclonus	-	Normal	MRI: normal	NA	[92]
	CJD	68	−	Dementia, Pyramidal or extrapyramidal disfunction, akinetic mutism	−	Normal	MRI: normal	NA	[95]
D178N	CJD-129M	71	NA	Progressive dementia, cerebellar signs, epilepsy	−	Slow waves	MRI: bilateral cortical atrophy	NA	[93]
	FFI	20s–60s	60% +	27 cases, dementia, myoclonus, visual and pyramidal issues	33% +	Negative	MRI: hyperintensities rare	NA	[92]
	CJD-129M	45	−	Memory loss, language impairment, no insomnia	+	NA	DWI: hyperintensities in cortex and basal	NA	[105]
	FFI	51	−	Memory personality changes, insomnia, hallucinations	NA	Atypical signa	DWI: high signal in frontal cortex and frontal lobes	NA	[125]
	FFI with CADASIL	58	+	Insomnia, movement impairment, mental dysfunctions, Co-existed with a Notch3 mutation	+	NA	MRI: white matter abnormality	Biopsy: spongiform changes, gliosis, prion deposits	[106]
	FFI	46	+	Autonomic nervous and cognitive dysfunctions, ataxia	NA	Extensive theta activity	MRI: frontal temporal lobe atrophy	NA	[126]
	FFI or CJD	27–60	4/7 +	7 cases of insomnia, FTD, anxiety, dementia	2/6 +	no PSWC	MRI: one patient typical CJD, one typical FTD others normal	All were 129MM	[107]
	FFI	57	+	Behavioral issues, rapidly progressive dementia, sleep disturbances	−	NA	MRI: frontal lobe atrophy	NA	[108]
	FFI	17–60	+	2 cases of sleep disturbances, psychiatric symptoms, ataxia	−	Slow waves	MRI: normal	NA	[127]
	CJD-129M	58	+	limb stiffness, bradykinesia, hypomimia, weight loss, no insomnia	NA	Diffuse slow-and sharp wave	DWI: hyperintensities in basal ganglia, frontal lobe cortices	NA	[104]
V180I	CJD	72	−	Bradykinesia, myoclonic jerks	−	Normal	DWI: high signal in several areas	NA	[109]
	CJD	70s	−	Dementia, pyramidal sign	−	PSWC	MRI: hyperintensities	NA	[92]
T183A	CJD	42	−	Dementia	−	Normal	Normal	NA	[99,110]
T188K	CJD	58	NA	Excessive daytime sleepiness, personality changes, motor impairment	+	Sharp waves	NA	NA	[111]
	CJD	40s–80s	37% +	16 cases, mostly dementia, but myoclonus or visual cerebellar disturbance	11/16 +	11/16 PSWC	MRI: 2/16 hyperintensities	NA	[92]
	CJD	39–76	1/7 +	8 cases of progressive dementia, pyramidal signs, Myoclonus and visual or cerebellar disturbances	6/8 +	1/8 PSD	MRI: various abnormalities	NA	[113]
	CJD	74	−	Cognitive impairment, ataxia, muscle weakness and myoclonus	+	PSWC	MRI: bilateral subcortical focal ischemic lesions rapid progressive brain atrophy	NA	[111]
	CJD	40–85	+ or −	30 cases of progressive dementia, motor symptoms, akinetic mutism	17/30 +	25/30 PSWC	DWI: cortical ribbon syndrome occurred frequently	NA	[112]
	CJD-hm	47	NA	Cerebellar issues, cognitive decline. visual disturbances	−	Normal	FDG PET: severe glucose hypometabolism	NA	[110]
E196K	CJD	71	NA	Rapid progressive dysfunctions of speech, memory, cognitive and physical movement	−	Normal	MRI: abnormal hyperintense lesions in bilateral corona radiate and centrum semioval	NA	[114]
E196A	CJD	60s–70s	−	2 cases of dementia, visual and cerebellar disturbance may be possible, pyramidal sign	+	PSWC	MRI: hyperintensities	NA	[92]
	CJD	76	−	Intellectual, mental and behavioral dysfunctions	+	PSD	MRI: ribbon-like high signal in bilateral cortices	NA	[115]
	CJD	55–76	−	3 cases of confusion, dystrophy, speak-and intelligence dysfunction	+	PSWC	MRI: high signals in bilateral frontal parietal lobes left occipital lobe or putamen	NA	[116]
	CJD	56	−	Progressive movement disorder, cognitive decline	+	No PSWC	DWI, MRI: high signal in several brain areas	Possible thyroid cancer	[118]
	CJD	42	−	Visual and psychotic symptoms	+	PSWC	DWI, MRI: ribbon-like high signal in several brain areas	NA	[117]
F198V	Probable AD	56	−	Memory loss, myoclonus hallucinations	NA	Diffused encephalic damage	MRI: moderate encephalic atrophy	NA	[93]
E200K	CJD	63	−	Sleep, memory and motor disturbances	+	Atypical waves	MRI: signal intensity in the caudate and putamen	NA	[119]
	CJD	40s–60s	1/9 +	9 cases of dementia, myoclonus, pyramidal signs	+	6/9 PSWC	MRI: 7/9 positive	NA	[92]
	CJD	42–71	4/30 +	27 cases of dementia, mental issues, myoclonus, pyramidal sign	20/27 +	13/30 PSWC	MRI: 26/30 abnormalities	NA	[121]
	CJD, FFI-like	51	−	Sleep disorder dizziness, paresis, ataxia, memory dysfunctions	−	Epileptiform discharges	MRI: normal	NA	[120]
E200A	CJD	60	−	Dementia, mental issues, myoclonus, pyramidal sign	+	PSWC	MRI: normal	NA	[92]
E200G	CJD	63	−	Pyramidal or extrapyramidal disfunction	NA	NA	NA	NA	[99]
V203I	CJD	80	−	Memory and language impairment, dizziness, blurred vision and ataxia.	+	Periodic activity	DWI: higher bilateral signal in frontal and parietal lobes	NA	[92,122]
R208C	Probable AD	81	NA	Hallucinations, cognitive decline	NA	NA	MRI: moderate atrophy	NA	[93]
R208H	CJD	40s–50s	1/2 −	2 cases, dementia, myoclonus, visual and cerebellar disturbance	1/2 +	NA	1/2 PSWC	NA	[92]
V210I	Panencephalitis and CJD	48	+	Motor clumsiness and hand dystonia, cognitive dysfunctions	NA	t-d background slowing and PSD	MRI: increased signal intensity at the basal ganglia and thalamus	NA	[123]
	CJD	59–69	−	3 cases of cognitive decline, dizziness, speech loss, sleep issues, hallucinations	+	Typical and untypical PSWC	MRI: ribbon-like high signals	NA	[124]
E219K	NA	NA	NA	NA	NA	NA	Possible risk modifier	NA	[92]
M232R	Genetic prion disease	44	NA	Behavioral changes, cognitive dysfunctions, ataxia	NA	NA	MRI: cerebral and cerebellar atrophy	NA	[93]

## 5. Genetic Modifiers in Genetic Prion Disease in Asia

Prion mutations, such as P102L, E200K, D178N, and V180I, could be related to diverse clinical phenotypes. Thus, genetic modifiers may change the disease course, resulting in earlier or later disease onset. Whether mutations are in the homozygous or heterozygous stage may also be an important factor. Although the majority of prion mutations are found in the heterozygous form, a few disease cases with homozygous forms have also been described. For example, M232R [78], Q212P [128], R136S [129] or E200D [129], V203I [73], and E200K [130,131] mutations been observed in the homozygous stage. Homozygous E200K and M232R may accelerate disease progression and may be related to younger disease onset compared to the heterozygous mutation [78,132]. However, no significant differences have been found between the clinical course and disease age of onset of the homozygous or heterozygous form of V203I [73] or T188K [110]. 

Concerning the D178N mutation, the most well-known genetic modifier is M129V. The modifier usually represents CJD and FFI in cases of the V/V and M/M genotype, respectively. However, a few atypical cases from Korea and Japan have been observed, where patients with the M/M genotype of D178N did not have insomnia. Additionally, this mutation is associated with highly variable disease phenotypes. It is possible that besides the M129 genotype, other genetic (or epigenetic) factors could also impact the D178N phenotypes. No reports are available supporting this view [20,23,103,107,111,133]. 

A Chinese study revealed a possible association between nonpathogenic E219K and GSS-related P102L mutations from patients with both mutations in comparison to patients with only P102L mutation. One patient harbored a heterozygous E219K mutation, while the others were homozygous E/E for codon 219. Patients with P102L-E219K had atypical disease phenotypes, such as rapid progressive dementia, PWSC on EEG, and short survival time (10 months). It may not be possible that E219K could impact survival time, since five other family members with P102L and without E219K also died in less than 2 years after disease onset. Other family members (2) were associated (with P102L and without E219K) with slow progressive dementia and longer survival (more than 40 months). Patient with E219K presented milder clinical in comparison to the GSS cases of the E/E genotype at codon 219. Further studies would be needed to determine how E219K could affect the disease course in the case of the P102L mutation. It may be interesting to explore additional genetic modifiers in this family, which may impact the disease course [96]. 

In a Japanese study describing a case of P102L-E219K and atypical GSS, no signs of PrP-positive plaques were observed in cerebral cortex sections stained with Congo red. The authors suggested that E219K may change the course of GSS in case of the P102L mutation, although other putative unidentified genetic or environmental factors might also affect the disease phenotype [40].

In addition to M129 or E219, additional possible genetic risk modifiers may exist. A few additional Asian studies have described potential genetic disease modifier factors of prion mutations. In 2016, a Korean study [26] analyzed disease-related genes in five CJD patients with V180I mutations. The authors suggests that prion diseases may share common pathways with other diseases, such as AD or PD. Several rare disease-related genes were identified, including LRRK2, LPA, FGF20, ACO1, and POSTN. Variants in these genes might affect the pathogenesis of CJD in patients with V180I [26]. Further studies are needed to verify these associations. Among the patients, one had an atypically long disease duration (death occurred 16.5 years after diagnosis). This Korean patient is the longest-known survivor of CJD. In most patients with V180I, the proteinase K-resistant isoform of prion protein is missing, resulting in a longer disease duration. However, the exact mechanisms underlying the extremely long disease duration remain unclear. It is possible that some variants of other genes may have neuroprotective effects in the case of V180I mutation. In one patient, a variant of ACO1 (rs189305274) was observed. ACO1 gene encodes the aconitase-1 protein, which is indirectly associated with AD and CJD [29]. It may regulate or inhibit processing of amyloid precursor protein; where the aggregation of proteins would be also important in prion diseases. ACO1 variants may also act as protective variants against prion-related neurodegeneration, and could delay disease progression [26,29]. 

Another Korean patient harboring the V180I mutation was found to have early onset AD [28]. The study data suggest that even though V180I may not directly impact AD progression, it could interact with other AD risk factors, such as variants in the ABCA7 and SORL1 genes. Their interaction could result in a reduced degree of neuroprotection and increased amyloid processing [28]. Risk modifiers have also been examined in CJD patients with E200K mutations. Diversities in phenotype, age of onset, and disease duration may be possible in patients with E200K. Furthermore, mutations also appear in asymptomatic individuals, suggesting that neuroprotective genetic modifiers may also be present in patients, such as Kallikrein B1 (KLKB1), Lysyl-TRNA Synthetase 1 (KARS), Neurexin 2 (NRXN2), and Laminin Subunit Alpha 3 (LAMA3) [26]. A study of E200K mutations in China investigated the relationship between disease phenotypes and the course of CJD [121]. Patients were also tested for Cytochrome P450 (CYP4X1) gene. An intronic variant rs9793471, which may modify the age of onset in cases of the E200K mutation. Of these patients, 29 carried the AA genotype, whereas one had the GA genotype. This study could not confirm whether CYP4X1 could be a risk modifier because of the small sample size [121]. 

Although all of these studies found promising data on risk modifiers in the case of genetic prion diseases, further analyses are needed to verify the role of these variants as risk modifiers. These data are preliminary and may be replicated in larger sample sizes or in other populations. 

## 6. Unique and Common Variants between Asian and Other Populations

Comparison of prion mutations in Asia and prion mutations worldwide reveals significant differences. A 10-year surveillance study performed in Japan analyzed 1685 patients, who were suspected to have prion disease [39]. Of these patients, 180 were categorized with definite prion disease, while 1026 and 97 cases were suggested as probable and possible prion disease cases, respectively. This study revealed that the incidence of CJD in Japanese patients was similar to that in European patients. In this study, 216 patients harbored prion mutations. The mutation pattern in Japan was different from that in the European Creutzfeldt–Jakob Disease Surveillance Network (EuroCJD). The most common Japanese mutations were V180I (41.2%), P102L (18.1%), E200K (17.1%) and M232R (15.3%). P102L (5.6% in EuroCJD) and P105L (not detected in EuroCJD) for GSS were significantly more common among Japanese patients than among other populations. However, other GSS-causing variants, such as A117V, have not been observed in Japanese (or other Asian) patients. Among CJD-related mutations, V180I (0.2% in EuroCJD) and M232R (not detected in EuroCJD) occurred significantly more frequently in Japan than in Europe. However, E200K (17.1% in Japan and 41.2% in Europe), V210I (0 and 16.2%, respectively), octapeptide insertion (1.4 and 9.9%, respectively), and D178N with either the 129M/M or V/V genotype [39] occurred less frequently in Japan than in European patients. A 2014 Korean study by Jeong et al. [134] revealed that the mutation pattern and prevalence of Korean genetic prion disease may be similar to those in Japanese patients. V180I, E200K, P102L, and M232R have been reported to be relatively common in Korean patients with genetic prion disease [134]. In 2015, the Chinese Surveillance Program analyzed 69 Chinese patients with prion disease. Chinese patients with CJD displayed a different pattern than that of Korean or Japanese patients. This study revealed that E200K and T188K are the most common mutations in Chinese genetic CJD. The prevalence of E200K in China is similar to that in Republic of Korea and Japan. FFI with D178N also seems to be a common mutation in China, as 39% of examined cases had this mutation [1,92]. To date, at least 30 prion cases of T188K have been reported in Chinese sporadic and familial CJD patients, but this mutation has rarely been observed in other populations. T188K appeared in two European cases (Austria and Germany). In addition, no reports are available on T188K in Asian populations (Korean or Japanese) other than Chinese [92,97]. E196A seems to be a rare mutation, and has been observed only in Chinese patients with sporadic CJD, but not in other populations in Republic of Korea and in Japan. However, another mutation in the same residue, E196K, seems to be more common among European patients, as it has been observed in only one Chinese case [92,97]. Interestingly, GSS cases with P105L were relatively frequent in Japan, but may be rare among Chinese individuals and have not yet been detected in Korean patients [39,98]. M232R and V180I have also been less frequent among Chinese patients, whereas they were dominant in Republic of Korea and Japan [1,97]. Figure 1d compares the differences between the most commonly occurring prion mutations in Korean, Japanese, and Chinese patients. The prevalence of M129V and E219K was also different between East Asia and Europe. In the general Korean and Japanese population, majority of individuals carry the MM allele for M129, while the heterozygous MV (approximately 6–7%) and the VV allele (less than 1%) may be rare. Meanwhile in European general populations, the MV and VV alleles quite common (35–51% and 8–12%, respectively). Heterozygous E219K appeared in approximately 7–8% of healthy Korean and Japanese populations, while it may be very rare among general European population [11]. 

Besides the common prion disease variants, several rare variants were reported among Korean, Chinese, and Japanese patients. These mutations were usually reported in a single case of disease. In Republic of Korea, a novel unique mutation, Y225C, was reported in an atypical CJD patient. Among the Japanese cases, two unique stop codon mutations (Y145X and Y162X) and a frameshift variant (D178fs) have been described. Among Chinese patients, several unique mutations have been identified, such as S17G, N97S, and R208C, which were observed in probable AD patients. However, these mutations were associated only with a single patient. Also, in the majority of them, no other affected family members were observed or the segregation could not be proven. As mentioned briefly in the introduction, these rare may not have strong evidence of pathogenicity [10].

## 7. Conclusions

This review summarizes prion mutations discovered in Asia. Several prion mutations have been identified in the Korean, Chinese, and Japanese populations. However, in other Asian countries, prion mutations have been poorly studied, and few studies have been conducted in Israel or India. As the number of patients with neurodegenerative diseases has been increasing, genetic analysis may be essential for accurate disease diagnosis. Next-generation sequencing provides much quicker and more cost-effective sequence analysis data. As there may be phenotypical similarities between prion diseases and other diseases, including *PRNP*, in gene panel studies, it is essential [1,135]. 

It is important to carry out genetic analysis of patients with prion diseases or possible prion diseases in Asian countries other than Republic of Korea, China, and Japan. Only a few reports describe prion mutations in other Asian countries, including India, Singapore or Taiwan. 

The first Southeast Asian case of fCJD with D178N-129V was observed in 2019 [136]. The age of onset ranged from to 42 to 65 years. All the affected family members developed progressive cognitive decline, myoclonus, ataxia, and personality changes. EEG revealed slowing, but triphasic periodic discharges were missing. In addition, the affected patients were negative for 14-3-3 in CSF. Interestingly, atypical symptoms were also observed. One family member had a more rapid disease course than the other family members with fCJD and D178N. Another family member developed insomnia, while the rest of the family did not. None of these family members can be analyzed for prion mutations [136]. The first case of Indian prion mutation was found in a large family with familial CJD [137]. Genetic testing revealed a D178N mutation (with homozygous valine at codon 129) in the two affected members. The proband developed memory dysfunction at 44 years of age with behavioral and language dysfunction. Within the next year, the patient became bed-bound, developed tremors and incontinences, and neglected personal hygiene. MRI revealed hyperintensities in basal ganglia and frontal temporal areas. EEG revealed non-specific slowing but not PSWC. Brain histology revealed cortical neuronal loss, spongiform changes, and gliosis in gray matter [137]. A second study described a 54-year-old patient with PRNP D178N from South India [138]. The patient developed sleep disturbances and rapid progressive cerebellar ataxia. Later, the patient developed language impairment, vocal cord palsy, and cortical visual deficits, but memory decline or myoclonus did not appear. EEG and CSF 14-3-3 levels were normal. Several of his family members developed similar clinical phenotypes between 41 and 67 years of age. One affected individual was heterozygous for the M129V mutation, but no homozygous V129 was detected in either the patients or unaffected individuals. The diagnosis of this family was atypical familial CJD [138]. One case of FFI was discovered in Singapore in 2018 in a 57-year-old female patient with a D178N mutation [139]. The patient’s family history was positive. MRI did not show any significant abnormalities, and EEG revealed loss of slow-wave activity. PET revealed typical FFI hallmarks, such as hypometabolism in the bilateral thalami and posterior cingulate cortex [139]. 

In Taiwan, a large family of individuals of Chinese ethnicity was reported to have a P102L mutation and transmissible spongiform encephalopathies. The age of onset was variable (27–64 years). Clinical diversity was observed among the affected relatives. Patients presented with ataxia as the initial disease symptom, followed by dementia. Disease progression was slow. One family member displayed personality changes, followed by ataxia with a rapid disease course. The other affected family members initially developed insomnia. No association was found between codons 129 and 219 variants and the clinical phenotype of GSS with P102L [140]. Also in Taiwan, the Creutzfeldt–Jakob Disease Surveillance Unit (CJDSU) investigated study on human prion diseases in Taiwan between 1996 and 2017. Genetic prion disease may be rare in Taiwan, since from 356 cases, only 8 carried genetic mutations: four of them carried P102L, two of them presented the E196A, had one patient had an octapeptide insertion (72 base pairs), and one of them carried the R148H mutation. All of them carried the homozygous methionine allele for residue 129. This study also analyzed the frequency of E219K and M129V variants. The heterozygous M129V and E219K (less than 2 and 3%, respectively) allele may be rare in both patients and healthy Taiwanese population [141].

Prion mutation patterns in Asia are different compared to mutations in Europe and the United States. For example, V180I and M232R are commonly detected in Korean and Japanese patients, but not in Chinese or other populations. T188K and E196A mutations are relatively common among Chinese patients. Asian prion mutations may show additional differences compared with prion mutations in Europe. For example, among GSS mutations, P102L and P105L mutations are common in Asians (especially Japanese patients). P102L was observed among European GSS patients, but less common than in Japan. Meanwhile, A117V-associated GSS has not yet been detected in Asia but is relatively common in the EuroCJD. In addition, V210I, V203I, or octapeptide substitutions were commonly detected among European CJD patients, while only a few Asian cases with these mutations have been reported [92]. 

In addition, it is important to study the pathogenic nature and phenotypic diversity of these mutations. For example, PRNP V180I and M232R have been identified in several typical or atypical CJD cases. Slow or rapid disease progression has been described in these mutations [1,39,75]. However, it has also been observed in patients who develop other forms of neurodegeneration, such as DLB [80], slow progressive ataxia [82], or dementia [75]. V180I is also common in Asia (especially in Republic of Korea and Japan), and was also observed in French patients [142]. Most cases were related to slow progressive late-onset CJD [85]. Additional phenotypes such as early onset AD [32] or CJD with AD-type pathology [57] may be possible. In 2012, Beck et al. [143] questioned the pathogenic nature of V180I and M232R, as they also appear in unaffected populations. In GnomAD, the frequency of M232R was 0.0008157 in the control populations and was only discovered in East Asian individuals. PRNP V180I was also found in GnomAD, with frequencies of 0.0003007 and 0.0002940 in the East Asian and South Asian populations, respectively. However, it was also found in Latin American individuals at a low frequency of 0.00005643 (https://gnomad.broadinstitute.org/, accessed on 10 November 2022). Because of 23andMe (https://www.23andme.com/en-int/, accessed on 10 November 2022), which is a genomics company, both failed to be categorized either as Mendelian variants or as benign variants. They were categorized as “variants with intermediate” penetrance variants, and their role in the disease should not be overlooked [10]. They still affect disease onset, especially with the combination of other genetic and environmental risk factors [28]. The cause of V180I and M232R dominance in Asian populations remains unclear. It is possible that racial and environmental factors could affect the presence of these mutations [143]. E200K is prevalent globally, and has been described in several Korean, Chinese, or Japanese patients [1]. CJD cases with V203I and V210I were also detected in patients globally, including Asian patients. However, they were more common in Europe, compared to Asia [1]. However, both V203I and V210I were found at low frequencies in unaffected populations (https://gnomad.broadinstitute.org/; accessed on 10 November 2022). Data from 23andMe does not indicate V203I is a highly penetrant variant, but could increase the risk of prion disease. Even though V210I was also found in cases with a positive family history of CJD, the ratio of familial cases with V210I may be low. V210I has been categorized as an “intermediate penetrant variant” that could be a strong risk factor for CJD [10].

Differences were found in Korean, Chinese, and Japanese prion mutation profiles compared with European patients. These differences in mutation profiles may be due to geographical isolation or an isolated ethnical group. Genetic screening for PRNP mutations in patients with neurodegenerative diseases should be essential [144,145,146,147,148]. Prion diseases with the same mutation may be associated with phenotypes related to genetic or epigenetic risk modifiers. M129V was verified as the main risk modifier; however, it may not be the only one. Currently, few studies have been performed in Asia on the impact of other genetic factors in the case of prion mutations, such as P102L, V180I, or E200K. For example, ACO1 may slow down disease progression in the case of the V180I mutation [26,27]. V180I interaction with AD risk factors, such as ABCA7 and SORL1 variants, may affect amyloid processing and the onset of AD or AD-like phenotypes [28,149]. E219K may result in an atypical disease course in the case of the P102L mutation [40]. In addition, several putative neuroprotective factors may affect the disease course of E200K related CJD, such as KLKB1, KARS, NRXN2, or LAMA3 [30]. However, these studies included only a few patients and need to be validated in larger patient groups. Further studies are needed to investigate the genetic factors that may affect the disease course and age of onset in cases of pathogenic PRNP mutations.

## Figures and Tables

**Figure 1 ijms-24-00625-f001:**
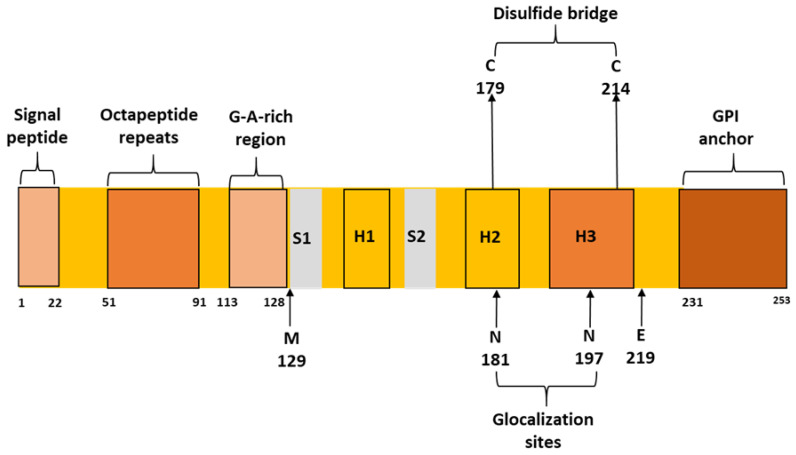
Schematic figure of PrP^C^ with the important domains and residues.

**Figure 2 ijms-24-00625-f002:**
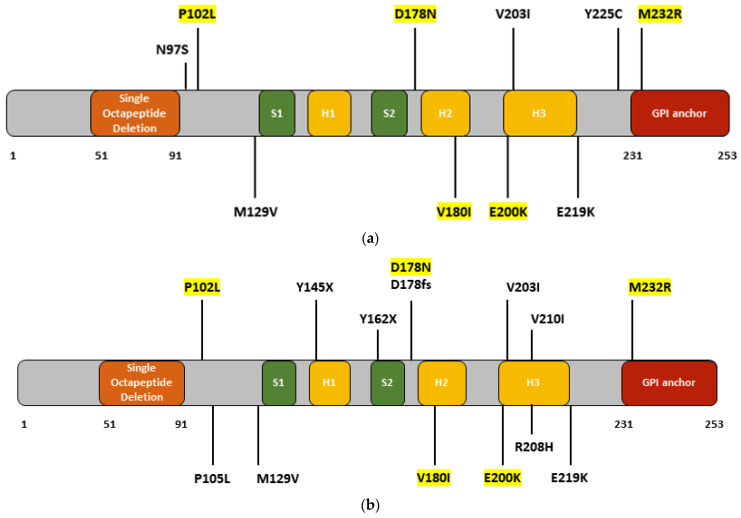
Mutations, reported in (**a**) Korean patients (**b**) Japanese patients (**c**) Chinese patients. Highlight mutations were mutually reported prion variants In Republic of Korea, Japan, and China. (**d**) Most commonly occurring mutations prion mutations in Republic of Korea, China, and Japan.

## Data Availability

Not applicable.

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
