# Peer review of "Prion Mutations in Republic of Republic of Korea, China, and Japan"

_ijms, 2022, doi:10.3390/ijms24010625_

Round 1

Reviewer 1 Report

The manuscript "Prion mutations in Asian countries" provides a very detailed description of prion mutations identified in Korea, China, and Japan. The tables and figures are particularly helpful in presenting the findings.

I would suggest renaming and reframing the manuscript as "Prion mutations in Korea, China, and Japan" given that these countries are the main focus. If desired, minimal information about other Asian countries (e.g., Israel) could be shared in the Discussion. I would also recommend attempting to involve, if an attempt has not already been made, counterparts from China and Japan as authors, who could perhaps add insight to the paper.

Minor comments:

The manuscript would benefit from further editing and proofreading (e.g., "ethnic" should be used rather than "ethical" in the Abstract, "...with a disease duration was 16.5 years" on Page 7 needs to be revised, "Prion mutations in China have been extensively studied for prion mutations," etc.

Be consistent with nomenclature (e.g., P102L is used throughout the paper, but Pro102Leu is used once as well.

Define "AOO" in the table heading.

Author Response

The manuscript "Prion mutations in Asian countries" provides a very detailed description of prion mutations identified in Korea, China, and Japan. The tables and figures are particularly helpful in presenting the findings.

I would suggest renaming and reframing the manuscript as "Prion mutations in Korea, China, and Japan" given that these countries are the main focus. If desired, minimal information about other Asian countries (e.g., Israel) could be shared in the Discussion. I would also recommend attempting to involve, if an attempt has not already been made, counterparts from China and Japan as authors, who could perhaps add insight to the paper.

Thank you very much, we moved the chapter from other countries to the Discussion section. In the future, we try to contact prion researchers from China and Japan, and we hope to collaborate with them.

Minor comments:

The manuscript would benefit from further editing and proofreading (e.g., "ethnic" should be used rather than "ethical" in the Abstract, "...with a disease duration was 16.5 years" on Page 7 needs to be revised, "Prion mutations in China have been extensively studied for prion mutations," etc

Thank you, errors have been fixed.

Be consistent with nomenclature (e.g., P102L is used throughout the paper, but Pro102Leu is used once as well.

Thank you, the error has been fixed

Define "AOO" in the table heading.

Thank you, “AOO” means an age of onset, I mentioned it in “Table 1”

Reviewer 2 Report

The manuscript by Kim et al provides a systematic review of reports on inherited prion diseases in three Asian countries, with a focus on how specific prion gene mutations present in carriers as well as how polymorphisms influence this presentation.

The article is divided into sections that initially treat Korea, Japan, and China separately

The authors are to be commended for what appears to be quite a comprehensive account of the literature.

Although a worthwhile effort, its descriptive nature makes for a bit of a tedious read. The authors attempt to mitigate the inherently archival nature of this report in several places throughout the manuscript by looking at broader trends and patterns. Still, the format of the article they chose, i.e., treating the countries separately in consecutive sections, is not in itslef conducive to conveying a bigger picture perspective.

Major comments:

- The tables in the article extend over several pages. This forces the reader to either memorize the table headers or to flick back and forth through pages. Moreover, the narrow widths of columns containing longer explanations contribute to the tables becoming unnecessarily long. To improve on this the presentation, this reviewer suggests to change the layout of the tables to landscape format and to optimize the width of columns with the objective to minimize the length of the tables.

- Each of the three tables is followed with an identical domain diagram of the prion gene, in which the country-specific mutations within the gene are highlighted. This reviewer suggests to combine these three figures into a single panel and add the latter at the top of the current Fig. 4. Doing so would greatly facilitate the recognition of differences and commonalities in the mutation landscape within the three countries, and would fit well with the content of the current Fig. 4, which could be a logical panel B in the same figure, because it compares the relative proportions of specific mutations across the three countries.

- The title of the article should be changed to 'Prion mutations in three Asian countries' since it doesn't cover large swaths of the Asian continent, including highly populated countries like India or Indonesia.

- The authors are encouraged to put additional efforts into distilling themes that emerge from their detailed descriptions.

Minor comments:

- This reviewer felt that in part the article is difficult to access due to a relatively large number of small writing errors. The authors are advised to have this aspect of the manuscript improved by recruiting the help of a native English speaking writer.

- Some of the errors are not writing errors but point toward a need to undertake more detailed error checks. An example of this shortcoming is the title of Figure 4, which currently has two instances of the word 'mutation' in it, one of which should be removed.

In summary, the authors are to be commended for a thorough listing of reports describing manifestations of familial prion diseases in three Asian countries. The article could be further strengthened by addressing the above points.

Author Response

The manuscript by Kim et al provides a systematic review of reports on inherited prion diseases in three Asian countries, with a focus on how specific prion gene mutations present in carriers as well as how polymorphisms influence this presentation.

The article is divided into sections that initially treat Korea, Japan, and China separately

The authors are to be commended for what appears to be quite a comprehensive account of the literature.

Although a worthwhile effort, its descriptive nature makes for a bit of a tedious read. The authors attempt to mitigate the inherently archival nature of this report in several places throughout the manuscript by looking at broader trends and patterns. Still, the format of the article they chose, i.e., treating the countries separately in consecutive sections, is not in itself conducive to conveying a bigger picture perspective.

Major comments:

- The tables in the article extend over several pages. This forces the reader to either memorize the table headers or to flick back and forth through pages. Moreover, the narrow widths of columns containing longer explanations contribute to the tables becoming unnecessarily long. To improve on this the presentation, this reviewer suggests to change the layout of the tables to landscape format and to optimize the width of columns with the objective to minimize the length of the tables.

We agree with this comment. We made the tables in landscape format and simplified them for better understanding.

- Each of the three tables is followed with an identical domain diagram of the prion gene, in which the country-specific mutations within the gene are highlighted. This reviewer suggests to combine these three figures into a single panel and add the latter at the top of the current Fig. 4. Doing so would greatly facilitate the recognition of differences and commonalities in the mutation landscape within the three countries, and would fit well with the content of the current Fig. 4, which could be a logical panel B in the same figure, because it compares the relative proportions of specific mutations across the three countries.

We fully agree on this comment, so we combined the four figures.

- The title of the article should be changed to 'Prion mutations in three Asian countries since it doesn't cover large swaths of the Asian continent, including highly populated countries like India or Indonesia.

Thank you. Reviewer 1 also had similar suggestion, so we changed the title to “"Prion mutations in Korea, China, and Japan”.

- The authors are encouraged to put additional efforts into distilling themes that emerge from their detailed descriptions.
Thank you, we discussed the clinical details, related to mutations in more in detail. We also added further description on disease progression if it was available.

Minor comments:

- This reviewer felt that in part the article is difficult to access due to a relatively large number of small writing errors. The authors are advised to have this aspect of the manuscript improved by recruiting the help of a native English-speaking writer.

Thank you, we tried to improve the language.

- Some of the errors are not writing errors but point toward a need to undertake more detailed error checks. An example of this shortcoming is the title of Figure 4, which currently has two instances of the word 'mutation' in it, one of which should be removed.

Thank you, the error has been fixed.

In summary, the authors are to be commended for a thorough listing of reports describing manifestations of familial prion diseases in three Asian countries. The article could be further strengthened by addressing the above points.

Thank you very much for the constructive comments.

Reviewer 3 Report

This review contains a comprehensive list of PRNP mutations found in patients diagnosed with a transmissible spongiform encephalopathy, as well as the clinical observations. The manuscript certainly warrants publication. There are two major issues that the authors should consider before publication. Firstly, there is no causality between mutations that are very rarely found and the disease of the patient, especially if there is no family history. The authors make this point at the end of the manuscript (lines 795-797) but this should be stated/explained much earlier. Secondly, in order to appreciate the unusual symptoms of some patients described by the authors it would be informative if the more usual progression of the diseases are given.

Minor comments.

Line 48 (and at several other places in the manuscript) I assume that with “normal prions” the authors

mean PrP in its non-infectious cellular form. When PrP is not infectious it is not in the prion configuration. Instead of using “normal prions” it would be better to use PrPc.

Line 51 PrPc should be defined.

Line 57 “Two post-translational proteins have been identified” The authors likely mean that PrPc is

trimmed both N and C-terminally? The protein is further post-translationally modified through glycosylation at two positions.

Paragraph starting at line 72. The descriptions of the time that it takes for the diseases to progress is

rather confusing. In line 74 the words rapid and short are used without indicating what is meant. Are the authors taking about hours, days, months, years? In

Line 77 the authors mention Parkinson’s disease-like features without explaining what these are.

Line 79 The authors don’t explain what amyloid is or what amyloid plaques are.

Line 87-88 It is unclear to me how mutations in PRNP are related to AD, FTD, and DLB? The relation

between these diseases and the TSE’s should be explained better.

Line 90 It is not explained what a-beta and tau protein are.

Line 94  Sentence needs to be restructured.

Line 101 the authors don’t explain why the listed mutations are considered pathogenic. Where and how

were these mutations identified?

Line 105 the M129V polymorphism has a long history. It deserves to be better described. E219K has

similar effects as the M129V polymorphism. This should also be described in more detail. The authors make it sound as if carriers of M129V are at a higher risk of getting TSE.

Line 112 It is not clear if any of the listed polymorphisms are disease associated or that they were found

in the general population.

Paragraph starting line 116 Is there any evidence that M129V and Ed219K or any other PRNP mutation is

associated with AD anywhere?

Line 122 What is the Korea Association Resource Group?

Line 129 What is “discriminated prion disease”?

Line 130 What is the frequency with which M129V and E219K are found in the Korean general

population? If this number is higher than the number listed in the text it would suggest a protective effect. The authors don’t mention this reported property of the M129V and E219K polymorphisms.

Line 136 The mutations listed are classically associated with certain TSEs. This should be explained.

Line 142 The authors should explain what a 14-3-3 signal is.

Line 189 Is it only with the D178N mutation that variable disease phenotypes are observed? This seems

like a trend of many cases listed in the manuscript.

Line 201 To appreciate the description of an atypical case the reader has to know what typical

progression and symptoms are.

Line 214 What AD or PD disease related genes do the authors have in mind?

Line 217 are the listed genes affecting AD and PD?

Line 225 Does amyloid peptide means A-beta?

Line 235 Is this the first E200K case ever reported or the first case in Korea?

Line 246 Is this the first case of V203I ever reported or the first case in Korea?

Line 253 Is this the first case of Y225C ever reported or the first case in Korea?

Line 263 Is this the first case of M232R ever reported or the first case in Korea? If it is the first case in

Korea maybe it could be mentioned where, and in what context, the mutation was first

detected?

Line 271 These individuals were from the control group?

Table 1  the meaning of the abbreviations (AOO and EEG) should be given.

Figure 1 Maybe the authors can provide a similar figure earlier in the manuscript listing the features of

PrP? These features are N-terminal signal peptide, C-terminal domain with the GPI anchor, indication of the repeat domains, glycosylation positions, disulfide bond. The domain that is folded in the cellular form could be shown. Also, the protective polymorphisms M129V and E219K could be indicated.

Line 318 The CJD Surveillance Committee of Japan?

Line 354 Did the patient have a M or a V in position 129?

Line 382 What effect the lack of PrP glycosylation has should be explained.

Line 393 I don’t think that it is explained what the symptoms of DLB are.

Line 402 What does “rarely” mean? Is it 2 cases globally? More?

Line 458 China is a big country. Is the analyzed sample size comparable to that of Japan and Korea?

Line 458 Mutations in PRNP.

Line 469 Finding one person with a mutation in PRNP does not show a causative link.

Line 507 Without a clear description of CJD systems the reasoning is not easy to verify.

Line 516 How big was the cohort?

Line 526 Was there a specific reason why M129V was examined?

Line 574 Just as a reminder where are these mutations commonly found?

Line 610 Is Lybia part of Asia?

General question, compared to those patients carrying mutations how many had a WT PRNP sequence?

Line 671 The observation that M129V is rare in patients agrees with the idea that those being

heterozygote have protection to acquiring a TSE! It would be interesting to know if the level of heterozygosity in the Asian population is similar to that of the European population.

Line 725 Was it explained what ACO1 is?

Line 742 Is it explained what the AA and GA genotype are?

Lines752-756 Japan – putative cases: 1685, confirmed cases: 180, PRNP mutations: 216. How many of

the confirmed cases had a mutation?

Line 771 Is this the total number of patients from which data is available? This is a low number relative

to the size of the country, as well as relative to studies in other countries.

Line 805 Last I checked Israel is not part of Asia.

Line 809 It is not clear to me what is essential.

Line 818 Are P102L and P105L mutations common in other geographical locations?

Line 837 It should be mentioned that 23andMe is a genomics company.

Line 839 Eliminate the second “variants” word.

Author Response

This review contains a comprehensive list of PRNP mutations found in patients diagnosed with a transmissible spongiform encephalopathy, as well as the clinical observations. The manuscript certainly warrants publication. There are two major issues that the authors should consider before publication.

Thank you very much for the constructive and detailed comments, we tried to revise the manuscript according to your suggestions.

Firstly, there is no causality between mutations that are very rarely found and the disease of the patient, especially if there is no family history. The authors make this point at the end of the manuscript (lines 795-797) but this should be stated/explained much earlier.

Thank you, we added this explanation to the introduction too.

Secondly, in order to appreciate the unusual symptoms of some patients described by the authors it would be informative if the more usual progression of the diseases are given.

Thank you, Reviewer 2 gave a similar suggestion. We discussed the clinical phenotypes, related to mutations more in detail. We also added further description on disease progression if it was available.

Minor comments.

Line 48 (and at several other places in the manuscript) I assume that with “normal prions” the authors mean PrP in its non-infectious cellular form. When PrP is not infectious it is not in the prion configuration. Instead of using “normal prions” it would be better to use PrPc.

Thank you, the issue has been fixed.

Line 51 PrPc should be defined.

Thank you, we added the definition: “cell-surface prion protein”

Line 57 “Two post-translational proteins have been identified” The authors likely mean that PrPc istrimmed both N and C-terminally? The protein is further post-translationally modified through glycosylation at two positions.

Thank you, we fixed this issue “Two post-translational cleavages of PrPC have been identified”… 

Paragraph starting at line 72. The descriptions of the time that it takes for the diseases to progress is rather confusing. In line 74 the words rapid and short are used without indicating what is meant. Are the authors taking about hours, days, months, years?

This sentence has been rewritten: “The disease duration may be variable; several GSS patients may die within a few months, while the could others survive for several (even more than 10) years.”

 In Line 77 the authors mention Parkinson’s disease-like features without explaining what these are.

This issue has been fixed: “The initial symptom of GSS may be ataxia, or Parkinsonism (such as tremor, bradykinesia, rigidity, and postural instability) and dementia may appear later in life”.

Line 79 The authors don’t explain what amyloid is or what amyloid plaques are.

“Amyloid plaques, which contain amyloid beta (Ab) peptide aggregates may appear in the brain”

Line 87-88 It is unclear to me how mutations in PRNP are related to AD, FTD, and DLB? The relation between these diseases and the TSE’s should be explained better.

This part has been rewritten: “Several patients with prion mutations were diagnosed with Alzheimer’s disease (AD), frontotemporal dementia (FTD), and dementia with Lewy bodies (DLB). Similarities have been observed between prion diseases and other neurodegenerative diseases (AD and FTD), since all of these diseases are associated with misfolded protein aggregation (such as Ab, microtubule-associated Tau protein). Also, PrPSc may accumulate together with Ab or tau protein.”

Line 90 It is not explained what a-beta and tau protein are.

Amyloid beta was explained earlier: “Amyloid plaques, which contain amyloid beta (Ab) peptide aggregates may appear in the brain”

Tau protein was explained earlier: “microtubule-associated Tau protein”.

Line 94  Sentence needs to be restructured.

“Furthermore, a few putative genetic disease modifier factors were also observed in these patients with prion mutations. We also will discuss the genetic modifier factors, described in Asian patients.”

Line 101 the authors don’t explain why the listed mutations are considered pathogenic. Where and how were these mutations identified?

We changed the “pathogenic” to “disease-related”

Line 105 the M129V polymorphism has a long history. It deserves to be better described. E219K has similar effects as the M129V polymorphism. This should also be described in more detail. The authors make it sound as if carriers of M129V are at a higher risk of getting TSE.

We discussed on M129V and E219K variants more in detail in the chapter on “Prion mutations in Korea”.

Line 112 It is not clear if any of the listed polymorphisms are disease associated or that they were found in the general population.

We mentioned in the text, they are probably benign variants, they were not reported in the Korean general population.

“. Furthermore, several possible benign missense or silent variants, such as D171S, A117A, G124G, and V161V, were missing in the Korean general population.”

Paragraph starting line 116 Is there any evidence that M129V and Ed219K or any other PRNP mutation is associated with AD anywhere?

Even though several studies analyzed their association with AD, the relation between E219K and M129V may be questionable. 

"

A 2007 paper by Jeong et al. [11] analyzed M129V and E219K in AD patients. The authors compared 276 patients with sporadic AD to 236 unaffected Korean individuals. No significant differences in the genotypes of codons 129/219 or their haplotypes were evident. These data suggest that codon 129/219 variants may not be directly associated with sporadic AD in Korea [11]. Similar data was detected among Japanese patients before [151]."

Line 122 What is the Korea Association Resource Group?

A small explanation was added: “KARE cohort study was established by Korea National Institutes of Health (KNIH), and they performed genome-wide association studies (GWAS) on a large community cohort. The goal of this study was to find genetic risk factors for diseases. The study by Lee et al. (2012) analyzed 22 patients with definite prion disease, 163 patients with suspected prion disease, and 296 individuals from KARE group (they were randomly selected).”

Line 129 What is “discriminated prion disease”?

We removed “discriminated” and changed it to “definite”

Line 130 What is the frequency with which M129V and E219K are found in the Korean general population? If this number is higher than the number listed in the text it would suggest a protective effect. The authors don’t mention this reported property of the M129V and E219K polymorphisms.

We changed this description: “The KARE study analyzed the frequency of the M129V and E219K variants in the three different groups. In the definite prion disease group, M/M and M/V allele ratio was 91% and 9%, respectively. The E/E and E/K allele ratios (for codon 219) were 94.5 and 5.5, respectively. Among the suspected CJD patients, the frequency of the M/V and E/K alleles was 4.91% and 6.79, respectively. In the KARE group, these respective frequencies of the M/V and E/K alleles were 5.8% and 7.82%, respectively. Furthermore, in the KARE group, one homozygous E219K mutation in the K/K allele was observed; this mutation may be very rare. This study found lower 129M/M and 219E/E allele frequencies in patients, compared to the studies by Jeong et al (2005) [12, 152]. However, the sample size of these studies was too small for precise analysis of these variants [12].”

Line 136 The mutations listed are classically associated with certain TSEs. This should be explained

We added “prion disease-associated mutation”

Line 142 The authors should explain what a 14-3-3 signal is.

We added a small explanation “The patient was positive for the 14-3-3 protein (14-3-3 is a signaling protein, CSF marker of prion diseases, including CJD)’.

Line 189 Is it only with the D178N mutation that variable disease phenotypes are observed? This seems like a trend of many cases listed in the manuscript.

This sentence has been removed for better understanding.

Line 201 To appreciate the description of an atypical case the reader has to know what typical progression and symptoms are.

We discussed the typical CJD/GSS/FFI symptoms/progression in the introduction.

“Genetic prion diseases can have diverse phenotypes, including Creutzfeldt–Jakob disease (CJD), fatal familial insomnia (FFI), and Gerstmann–Sträussler–Scheinker disease (GSS) [8]. Typical familial CJD is associated with rapid disease progression, short survival time (less than a year), and progressive dementia with motor dysfunctions (myoclonus, tremor). In the brain, PrPSc plaques may be associated with gliosis and neuronal loss. The initial symptom of GSS may be ataxia, or Parkinsonism (such as tremor, bradykinesia, rigidity, and postural instability) and dementia may appear later in life. The disease duration may be variable; several GSS patients may die within a year, while the majority of patients may survive for several (even more than 10) years. Amyloid plaques, which contain amyloid beta (Ab) peptide aggregates may appear in the brain, particularly in the cerebellum. The initial symptoms of FFI usually include insomnia and dysautonomia, followed by motor and cognitive impairments in later disease stages. Disease duration may be relatively short; patients can die less than 2 years after disease onset. FFI neuropathy may be diverse and includes loss of thalamic nerves, thalamic atrophy, inferior olivary nucleus atrophy, or PrPSc deposition in the midbrain or hypothalamus [9]. All three genetic prion diseases may represent atypical forms of the disease with different symptoms and longer or shorter disease durations. Furthermore, atypical disease phenotypes may also be related to genetic prion mutations.”

Line 214 What AD or PD disease related genes do the authors have in mind?

“Genes, involved in this study included AD risk genes (for example Aconitase 1 or ACO1; Lipoprotein A or LPA; Periostin or POSTN; Structural maintenance of chromosomes pro-tein 5 or SMC5) or PD risk genes (Fibroblast Growth Factor 20 or FGF20; leucine-rich repeat kinase 2 or LRRK2; 2-Hydroxyacyl-CoA Lyase 1 or HACL1 ).’

Line 217 are the listed genes affecting AD and PD?

They are risk factor for Alzheimer’s disease or Parkinson’s disease

Line 225 Does amyloid peptide means A-beta?

Yes, we fixed it to Ab

Line 235 Is this the first E200K case ever reported or the first case in Korea?

Yes, this is the first case, we mentioned it.

Line 246 Is this the first case of V203I ever reported or the first case in Korea?

We included it into the text.

Line 253 Is this the first case of Y225C ever reported or the first case in Korea?

This was the first case reported ever. We included in the text.

Line 263 Is this the first case of M232R ever reported or the first case in Korea? If it is the first case in Korea maybe it could be mentioned where, and in what context, the mutation was first detected?

We included it in the text.

Line 271 These individuals were from the control group?

We included in the text.

Table 1  the meaning of the abbreviations (AOO and EEG) should be given.

We added the meaning of abbreviations: AOO means the age at onset, EEG means electroencephalography)

Figure 1 Maybe the authors can provide a similar figure earlier in the manuscript listing the features ofPrP? These features are N-terminal signal peptide, C-terminal domain with the GPI anchor, indication of the repeat domains, glycosylation positions, disulfide bond. The domain that is folded in the cellular form could be shown. Also, the protective polymorphisms M129V and E219K could be indicated.

Thank you, a schematic figure has been added to the introduction.

"Figure 1 shows a schematic structure of prion protein with important positions and residues. The polymorph methionine 129 and glutamic acid 219 residues are also included."

Line 318 The CJD Surveillance Committee of Japan?

“Octapeptide insertions were also mentioned by the CJD Surveillance Committee of Japan, which performed an extensive study on patients with prion disease. This 10-year long study was established in 1999, and they performed prospective surveillance on human prion diseases. Experts, involved in this study collected data (including imaging, EEG, genetic, biomarker, or neuropathology) from patients. Patients, which were suspected of having prion disease were investigated by this study.”

Line 354 Did the patient have a M or a V in position 129?

“The patient harbored the MM homozygous form of M129V and presented with rigidity, speech disturbance, paranoia, and seizures at 60-years-of-age. CSF was positive for 14-3-3 protein and EEG revealed slow bilateral sharp complexes. This study revealed that MM and VV genotypes in PRNP codon 129 may increase the risk for multiple system atrophy (MSA) in patients with PD”

Line 382 What effect the lack of PrP glycosylation has should be explained.

We briefly mentioned it in the introduction: “Two glycosylation sites were identified in PrPC (N181 and N197), which are occasionally occupied by complex N-glycans. These residues were suggested to play a crucial role in neuroprotection and in the prevention of protein assembly and toxicity”

Line 393 I don’t think that it is explained what the symptoms of DLB are.

“Patient was suspected of having DLB since he developed hallucinations, Parkinsonism, but symptoms progressed more rapidly than in typical DLB”

Line 402 What does “rarely” mean? Is it 2 cases globally? More?

This sentence has been changed: “The majority of cases with V180I in Japan featured a negative family history, but the familial form of V180I but a few cases with positive family history were also observed.

Line 458 China is a big country. Is the analyzed sample size comparable to that of Japan and Korea?

This question may be difficult to answer. We added a short explanation

“Several studies are available on Chinese patients with prion disease and mutations. However, it may be possible, that there are several additional patients, who remained undiagnosed for prion disease, especially from undeveloped areas of the country.”

Line 458 Mutations in PRNP.

This error has been fixed.

Line 469 Finding one person with a mutation in PRNP does not show a causative link.

We added an explanation: “It may be unclear, whether the PRNP S17G could impact the disease phenotypes.”

Also, we mentioned this issue in the introduction:

‘]. Besides the common prion disease-related variants in Asian patients (such as P102L, V180I, E200K, M232R), several unique rare variants have appeared in Korean, Chinese, and Japanese patients, which were reported only in a single patient (will be discussed later). The majority of rare mutations may not have strong evidence of pathogenicity, especially if they do not have any family history of disease or segregation cannot be proven. However, their rule in disease may not be ruled out, especially if they were not observed, or were very rarely observed in reference databases, such as GnomAD [134].”

Line 507 Without a clear description of CJD systems the reasoning is not easy to verify.

“Several patients display sporadic CJD-like (confusion, depression, memory issues) symptoms.”

Line 516 How big was the cohort?

“G114V has been reported in familial prion disease, which occurs in patients in their 40s or 50s. A large family with 49 members (including spouses) was analyzed with this mutation. Proband developed progressive dementia at the age of 45, and also experienced tiredness, lethargy, and sleep issues. She also had motor issues, such as myoclonus, Babinski sings or hyperreflexia. MRI revealed bilateral atrophy in different brain areas. In this family, other members with similar mutations were found to be carriers of G114V, including the two siblings of the proband. Also, one relative (son of her first cousin) experienced progressive memory impairment and ataxia at the age of 32”

Line 526 Was there a specific reason why M129V was examined?

“The V allele of M129V has been examined in patients with mesial temporal lobe epilepsy (MTLE). In 2007, a similar study reported that M129V may be a possible risk modifier for MTLE in the Italian population. However, the Chinese study could not confirm the association [99].”

Line 574 Just as a reminder where are these mutations commonly found?

We removed that sentence. However, several mutations, such as E200G, E200A, R208C, V210I, and M232R were relatively rare in China.

Line 610 Is Lybia part of Asia?

 We removed this part since these patients were African immigrants.

General question, compared to those patients carrying mutations how many had a WT PRNP sequence?

We mentioned this issue in the introduction: ‘A small percentage (10-15%) of prion diseases may be related to genetic mutations in the prion gene (PRNP)’.

Line 671 The observation that M129V is rare in patients agrees with the idea that those being heterozygote have protection to acquiring a TSE! It would be interesting to know if the level of heterozygosity in the Asian population is similar to that of the European population.

Thank you, we added a short paragraph in the chapter “Prion mutations in Korea”.

“The prevalence of M129V and E219K was also different between East Asia and Europe. In the general Korean and Japanese population, majority of individuals carry the MM allele for M129, while the heterozygous MV (approximately 6-7%) and the VV allele (less than 1%) may be rare. Meanwhile in European general populations, the MV and VV alleles quite common (35-51% and 8-12%, respectively). Heterozygous E219K appeared in approximately 7-8% of healthy Korean and Japanese populations, while it may be very rare among the general European population [10].”

Line 725 Was it explained what ACO1 is?

We explained earlier. ACO1 is gene for Aconitase 1 protein.

“ACO1 gene encodes the aconitase-1 protein, which is indirectly associated with AD and CJD.”

Line 742 Is it explained what the AA and GA genotype are?

The rs9793471 variant is an intronic variant, guanine to adenine exchange. AA allele means that these patients were homozygous for the alternative allele. The GA genotype means the patient carried the heterozygous allele.

Lines752-756 Japan – putative cases: 1685, confirmed cases: 180, PRNP mutations: 216. How many of the confirmed cases had a mutation?

“Of these patients, 180 were categorized with definite prion disease, while 1026 and 97 cases were suggested as probable and possible prion disease cases, respectively.”

Line 771 Is this the total number of patients from which data is available? This is a low number relative to the size of the country, as well as relative to studies in other countries.

We agree with this comment. There may be much more patients with prion disease in China. As we mentioned earlier, in the chapter “Prion mutations in China” there may be several undiagnosed cases in China.

Line 805 Last I checked Israel is not part of Asia.

I think Israel is actually part of West Asia. However, since the affected patients are African immigrants from Lybia, we removed this paragraph.

Line 809 It is not clear to me what is essential.

The sentence (and the paragraph) were removed.

Line 818 Are P102L and P105L mutations common in other geographical locations?

They seem to occur more frequently in Japan, compared to other populations.  P102L was detected in European patients, but less frequently than in Japan.

Line 837 It should be mentioned that 23andMe is a genomics company.

Thank you, we mentioned it.

Line 839 Eliminate the second “variants” word.

Thank you, it has been removed.